**A dedicated robust instrument for water vapor generation at low-humidity for use with a laser water**
**isotope analyzer in cold and dry polar regions.**
Christophe Leroy-Dos Santos[1], Mathieu Casado[1,2], Frédéric Prié[1], Olivier Jossoud[1], Erik Kerstel[3], Morgane
Farradèche[1], Samir Kassi[3], Elise Fourré[1], Amaëlle Landais[1, *]
[1] Laboratoire des Sciences du Climat et de l'Environnement, CEA-CNRS-UVSQ-Paris Saclay-IPSL, Gif-sur-
Yvette, France
[2] Alfred Wegener Institut, Helmholtz Center for Polar and Marine Research, Potsdam, Germany
[3] Laboratoire Interdisciplinaire de Physique, CNRS - Université Grenoble Alpes, Grenoble, France
[*] corresponding author: amaelle.landais@lsce.ipsl.fr
**Abstract**
Obtaining precise continuous measurements of water vapor isotopic composition in dry places (polar or
high-altitude regions) is an important challenge. The current limitation is the strong influence of humidity
on the measured water isotopic composition by laser spectroscopy instruments for low-humidity levels
(below 3,000 ppmv). This problem is addressed by determining the relationships between humidity and
measured $\delta^{18}O$ and $\delta D$ of known water standards. We present here the development of a robust field
instrument able to generate water vapor, down to 70 ppmv, at very stable humidity levels (average $1\sigma$
lower than 10 ppmv). This instrument, operated by a Raspberry interface, can be coupled to a commercial
laser spectroscopy instrument. We checked the stability of the system as well as its accuracy when
expressing the measured isotopic composition of water vapor on the VSMOW-VSLAP scale. It proved to
be highly stable during autonomous operation over more than one year at the East Antarctic Concordia
and Dumont d'Urville stations.


27       **1.  Introduction**

The recent development of laser spectroscopy instruments now enables the continuous measurement of
the isotopic composition of water vapor at many observation stations all around the world (Bailey et al.,
2015; Bastrikov et al., 2014; Schmidt et al., 2010; Sodemann et al., 2017; Tremoy et al., 2011). In particular,
the isotopic composition of the water vapor has proven to be a very useful tool to document moist synoptic
events in many locations (Bonne et al., 2014; Guilpart et al., 2017). In polar regions, the water vapor
isotopic signal is not only useful to detect the origin of moist air (Bréant et al., 2019; Kopec et al., 2014)
but also to improve the interpretation of the isotopic composition of water in surface snow and ice core
archives (Steen-Larsen et al., 2014). Indeed, exchanges are occurring after deposition between the surface
snow and the water vapor leading to modifications of the isotopic composition of the former and hence
of the archived ice (Casado et al., 2016, 2018; Ritter et al., 2016).
Obtaining continuous and accurate measurements of the water vapor isotopic composition expressed on
the VSMOW-VSLAP scale measurements of the water vapor isotopic composition at Concordia station in
central Antarctica is a key scientific challenge since the deep ice core drilled there, EPICA Dome C, provides
the oldest continuous water isotopic record expressed on the VSMOW-VSLAP scale to date (Jouzel et al.,
2007). It is thus a key reference for the study of past climate, and a correct interpretation of the isotopic
record relies on the quantification of the transfer function between climate parameters and water isotopic
composition in ice, itself influenced by exchanges with water vapor in the upper layers of the firn (Casado
et al., 2018). Such knowledge is also of uttermost importance for the interpretation of water isotope
records from the starting deep drilling project "Beyond EPICA-Oldest Ice" (https://www.beyondepica.eu),
whose aim is to drill a 1.5-million-year old ice core at the Little Dome C site located 40 km away from
Concordia station, hence with similar low temperature and humidity conditions.
One of the main limitations of the current commercial instruments when deployed in polar regions is their
relatively poor performance at low water vapor concentration. Generally, the precision of the measured
isotopic ratios $\delta^{18}O$ and $\delta D$ rapidly worsens when the water mixing ratio decreases to humidity levels
below 3,000-5,000 ppmv (part-per-million per volume) (Bonne et al., 2014; Weng et al., 2020). However,
in remote continental areas in Greenland and Antarctica, temperatures in winter can drop to very low
values, leading to humidity levels down to 10 ppmv (Genthon et al., 2017). Arguably one of the most
extreme experiments for continuous measurement of the water vapor isotopic composition was the
deployment of a commercial Picarro L2130-i instrument at the East Antarctic French-Italian station of
Concordia where the mean annual temperature is around -54°C and the humidity barely exceeds 1,000
ppmv during the warmest summer days (Casado et al., 2016). For such applications, there are two major
impacts of low-humidity on the raw isotopic signal: first, we generally observe an apparent increase in the
$\delta^{18}O$ and $\delta D$ with decreasing humidity level and second, the standard deviation associated with the
continuous measurements of $\delta^{18}O$ and $\delta D$ of the water vapor increases. This can lead to overall
uncertainties of several ‰ for $\delta^{18}O$ and tens of ‰ for $\delta D$. It is thus of uttermost importance to have a
correct determination of the humidity dependency of the water vapor isotopic ratios.
Commercial instruments from Picarro Inc. are usually associated with a Picarro Standard Delivery Module
(SDM) designed to generate humidity at stable levels between 5,000 and 30,000 ppmv. Using such a set-
up for humidity levels below 5,000 ppmv leads to large uncertainties in the determination of the humidity
influence on the water vapor isotopic composition (e.g. Guilpart et al., 2017). These uncertainties are due
both to the instability of the water vapor generation using the SDM (in terms of water concentration –
humidity — and/or isotopic composition) and to the analytical noise in the spectroscopy measurements
when the absorption signals are weak. An alternative commercial device is the LGR (Los Gatos Research)

calibration system (Water Vapor Isotope Standard Source, WVISS), which uses a nebulizer to instantaneously evaporate micro-droplets of liquid water from a standard reservoir into a large (1 L) vaporizing chamber (Dong and Baer, 2010). This system is very stable and well adapted for a humidity range between 2,500 and 25,000 ppmv (Aemisegger et al., 2012).

Several home-made water vapor injection systems have been developed with the specific aim to achieve a better stability of the generated humidity at low-humidity levels. A first approach is to use a dew point generator injecting small amounts of water into dry air (Lee et al., 2005; Wang et al., 2009). This approach is time consuming as it takes long to reach equilibrium and relies on a very precise knowledge of the temperature to quantify the isotopic fractionation. A method using a piezoelectric microdroplet generator into a dry air stream could generate water mixing ratios between 12 and 3,500 ppmv (Iannone et al., 2009; Sturm and Knohl, 2009; Sayres et al., 2009). However, adjustment of humidity level and long-term stability were difficult to obtain with such devices. Systems relying on the use of syringe pumps were also built by Gkinis et al. (2010) and Tremoy et al. (2011): a small fraction of the input stream of liquid water is introduced into a hot oven where water is vaporized in the presence of a dry air flow. These systems cover humidity range between 2,000 and 30,000 ppmv. Finally, bubbler systems, in which dry air flows through a large volume of water to create saturated vapor, are very robust but can only produce water vapor at high-humidity levels (Ellehoj et al., 2013). The aforementioned devices are unfortunately not well suited for automatic long-term operation at low-humidity levels. During the 2014-2015 summer field season at Concordia station in Antarctica, a home-made humidity generator specifically designed for low-humidity levels (Landsberg, 2014) has been deployed (Casado et al., 2016). The device used dual high-precision, low-volume, syringe pumps to generate stable humidity levels at two different isotopic compositions over the range from 100 to 800 ppmv (Casado et al., 2016). Unfortunately, we observed quite a large scattering among the isotopic values measured at similar humidity levels, as well as a large discrepancy between the humidity dependency of the water isotopic ratios measured in the field and the one measured in the laboratory. Upon return to the laboratory, these defaults were traced primarily to tiny leaks in the water supply lines to the syringes.

Therefore, we re-engineered the prototype by Landsberg (2014) in order to develop a robust and autonomous device for stable low-level humidity generation for the purpose of precise humidity calibration of spectroscopic instruments. Such devices have now been operating with minimum manual intervention for more than one year at two polar stations in Antarctica, Dumont d'Urville and Concordia, coupled to Picarro laser spectroscopy instruments. We detail here the technical description of the instrument and show key performance characteristics, enabling, for instance, a discussion of small amplitude signals such as the diurnal variability of the water vapor isotopic composition in remote dry sites in East Antarctica.

**2. New vapor generator for low-humidity levels**

The low-humidity level generator (LHLG) developed here relies on the same principle as the one developed
by Landsberg (2014), i.e., a steady, undersaturated evaporation of a liquid water droplet at the tip of a
needle into a dry air stream inside a small evaporation chamber. Based on this first prototype, the
instrument has been remodeled including a specific hardware and software design.

**2-1-Physical principle**
The LHLG is based on undersaturated evaporation of a small droplet at the tip of a needle (Figure 1). Liquid
water is pushed through a needle around which dry air is flowing. Dry air is obtained from a bottle of high
purity synthetic air with pressure regulation through two manometers connected in series. The mass flux
of water $f_L$ is kept low compared to the air mass flow $f_A$ so that the relative humidity $RH$ of the downstream
moist air flow remains low ($RH < 0.1$). Therefore, the air stays largely undersaturated and its humidity is
controlled only by the flow of liquid water in the needle and that of the dry air upstream of it. The mixing
ratio (or humidity) of the air as classically provided by a Picarro instrument is given by:

$$MR = \frac{d_{H2O} \times f_L \times R \times T_{st}}{f_A \times P_{st} \times M_{H2O}} \quad \text{(eq. 1)}$$

where $d_{H2O}$ = 1000 kg m$^{-3}$ is the density of water, R = 8.314 J mol$^{-1}$ K$^{-1}$ is the universal gas constant, $T_{st}$ =
293.15 K and $P_{st}$ = 1013.25 hPa are standard conditions of temperature and pressure and $M_{H2O}$ = 18.10$^{-3}$
kg mol$^{-1}$.

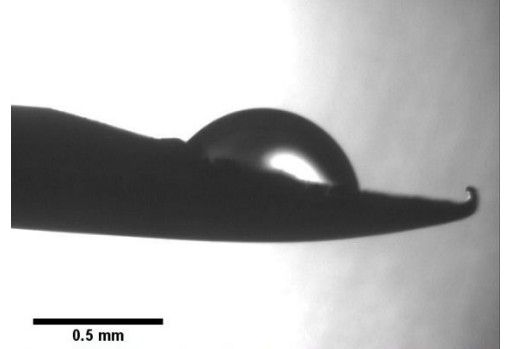 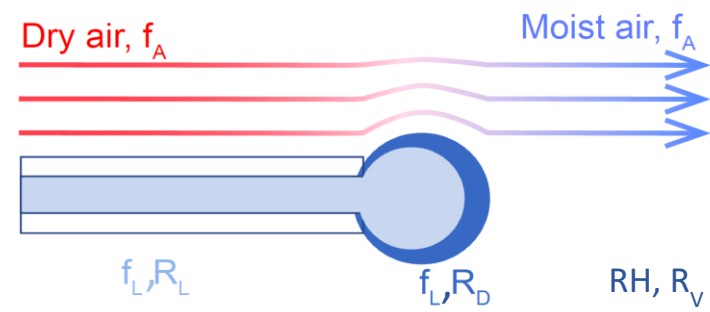


**_Figure 1_**: *Evaporation of a droplet in the humidity generator chamber: left, picture from the prototype from*
*Landsberg (2014); right, schematics of the water molecules being transferred to the air flow (Casado,*
*2016).*

Physically, when the flux of water or air is changed, there is first a transient regime during which the radius
of the droplet changes, modifying the evaporative surface and therefore the humidity of the outgoing air.
Once a stationary regime is reached, the radius of the droplet is stabilized and the humidity is given by
equation 1. In this regime, there is no accumulation of water molecules in the system and therefore the
isotopic composition of the vapor produced is equal to the isotopic composition of the liquid water
injected in the needle: $R_V = R_L$ (note that because of the fractionation during the transition phase, the
isotopic composition of the droplet $R_D$ is different from $R_L$ and $R_V$, see Kerstel, 2020). When changing the
flux of evaporating water, we modify the size of the evaporating surface and therefore the radius of the
drop. The evolution of the radius of the drop can be obtained from the resolution of a non-linear
differential equation of the volume V of the drop:

$$dV/dt = f_L - f_{evap} \qquad \text{(eq. 2)}$$

where $f_{evap} = k_e \times S$ is the evaporation flux depending of $k_e$, the evaporation rate, and $S$, the surface area of
the drop exposed to the dry air. A good approximation is to consider the shape of the drop as a fraction of
a sphere of variable radius intercepted by the surface of a disk of constant radius (the syringe tip). By
solving numerically the differential equation (2), it is possible to faithfully simulate the behavior of the
device under changing conditions (Kerstel, 2020). This numerical approach validates the theoretical
explanation of the undersaturated evaporation of the droplet. Importantly, it is noted that in steady-state
as is the case for our application, the isotopic composition of the generated humid air is identical to that
of the injected water stream, and therefore does not depend on the infusion rate, nor on the specific
humidity.

**2-2- Instrument conception**

-  **Technical realization**
As the LHLG relies on operating in a stationary regime, it is important that the dry air input and the water
input are steady. Thus, the air and water fluxes, as well as the air pressure in the evaporation chamber are
controlled by electronic PID regulators. Temperature intervenes through its effect on fractionation and
the evaporation rate (apart from a negligible effect on the flow controller stability), which could lead to a
departure from steady-state operation. For these reasons, the temperature of the evaporation chambers
was maintained at 20°C (within 1°C over 24 hours).
The dry air flux is regulated by a high-precision mass flow controller (Vögtlin GSC-A9TS-DD22), that has an
operating range from 6 to 600 sccm (std cm$^3$ min$^{-1}$) and an accuracy of 3.3 sccm. The water flux is regulated
by a high-precision syringe pump (Harvard Apparatus Pump 11 Pico Plus Elite Dual), which can produce a
water flow down to 10.8 pL min$^{-1}$ with an accuracy of 0.35 % using syringes with a volume ranging from
10 µL to 250 µL. We operate in the routine mode with a dry air flow of 300 sccm and a water flow between
0.02 to 0.5 µL min$^{-1}$ using mainly 50 or 100 µL syringes. A syringe pump is equipped with two syringes that
provide two water flows into two evaporation chambers in parallel (Figure 2). Each syringe is connected
to a water reservoir and to an evaporation chamber by a double 3-way liquid valve (Rheodyne MXX777603)
switching from an "infuse" mode to a "withdraw" mode to refill the syringes. The water in the water
reservoirs is sampled every month to check its isotopic composition and renewed when the level of water
is below half the maximum level. A maximum evolution of the isotopic composition of the lab-standard
filling the water reservoirs has been observed as 0.05‰ and 0.5‰ respectively for $\delta^{18}O$ and $\delta D$ over a 2-
month period.
A major change to the instrument designed by Landsberg (2014) is the introduction of the double 3-way
valve with leak-tight connections and an internal volume of 1.9 µL. This modification is an important
improvement as it enables automatic handling of the lab-standards from a reservoir to the evaporation
chamber with a robust connection, avoiding in particular potential air bubbles in the water flow. Indeed,
the compressibility of air bubbles trapped in the water flow can lead to flow irregularities by amplification
of small non-linearities in the progression of the syringe plunger. This would lead to non-steady state
operation, which in turn would create artefacts in the humidity and isotopic composition, reducing the
performance of the calibration device (see Kerstel 2020). In addition, the 3-way valve provides the
opportunity of a "withdrawn" mode in which the syringes draw lab-standard water from a reservoir. When
equipped with 100-µL syringes, the instrument can operate for several hours up to one day between refills.
With the addition of the auto-refill option and the effective suppression of bubbles, the instrument can be
used unattended for many months, as required for an Antarctic winter field campaign.

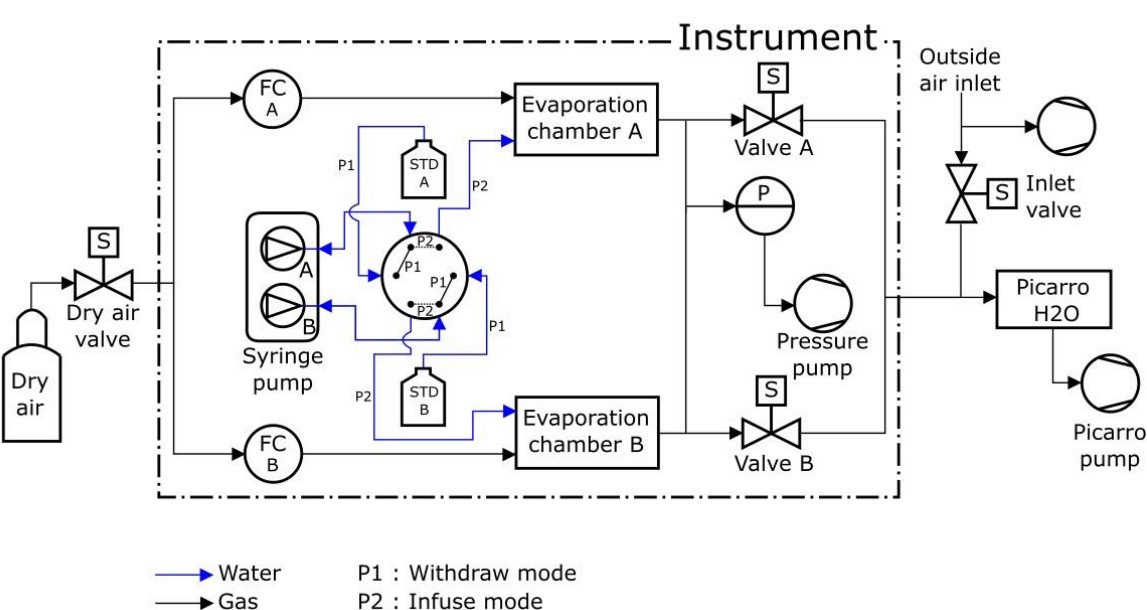


*__Figure 2__: Humidity generator schematic diagram (see supplementary Table S1 for details on the different elements)*

The evaporation chambers are stainless steel cylinders equipped with specific connectors (Swagelok Ultra-Torr SS-4CD-TW-25) holding silicon rubber septa through which needles are inserted toward the middle of the chamber. The pressure in both chambers is regulated by a pressure controller (Bronkhorst P-702CV-1K1A-AAD-22-V) with a precision of 3 mbar in a range from 0 to 1,000 mbar. This pressurization of the two chambers combined with the relatively high flow (higher than required by the infrared spectrometers) enables maintaining a steady state whether or not the infrared spectrometer is connected, and increases the time efficiency of calibration procedures. The spectrometer is not sensitive to the inlet pressure, the precision of the pressure controller is not an essential aspect. On the contrary, the precision of the flow controller is key for the precision of the humidity level produced by the instrument: it is of 1% for the air flow which is comparable to the precision of the measurement of the humidity level with the optical spectrometer. When the instrument is connected to the infrared spectrometer, the excess humid air flow is exhausted to the room through the pressure pump and the spectrometer only pumps what is required (Figure 2).

The control of the instrument is ensured by a Raspberry Pi that can be interfaced to a Picarro water analyzer (L2130-i in our case) in sequencer mode (see below). The hardware has been designed to meet the specifications dictated by field conditions: 1) All components are fixed in a transportable case (except the dry air bottle), isolated from vibration by an anti-vibration foam. 2) A panel of connectors (HDMI, USB, Ethernet, etc.) ensures the accessibility to the instrument when it is closed. 3) The electrical and electronic parts (e.g. power supply, Raspberry Pi) are separated from the rest of the instrument (e.g. sensors, gauges). Both the electrical and electronic parts are fully and easily accessible in case of failure.

- **Software details**

The control software has been developed using open source Python libraries and homemade drivers, including a user interface displaying the state of relevant components and the value of the different sensors. The software (HumGen) can be downloaded on line (https://github.com/ojsd/humgen; https://doi.org/10.5281/zenodo.4003465).

The LHLG can operate in eight different states, each state representing a specific setup for each element (valves position, syringe pump infusion rate, dry air flow rate, pressure). Those eight states can be divided into three categories: a routine mode, an expert mode and a humidity dependence calibration mode. The simple mode is composed of six predefined states referring to the classic isotopic calibration in everyday routine operation (Table 1): 1) measurement of the outside air water vapor isotopic composition; 2) drying of the cavities; 3) "humidity boost", in order to reach faster the desired humidity level in the cavities; 4)

injection of the standard A in the corresponding evaporation chamber at a set humidity level; 5) injection
of the standard B in the corresponding evaporation chamber; 6) refill of the syringes. The expert mode is
useful to adjust each parameter manually: flow rates on the controllers FCA and FCB, opening of the
electrovalves A and B, mode (infuse or withdraw) and infused rate for the syringe pump, pressure
regulation, state of the double three-way valve, activation of the pressure pump at the exhaust, opening
of external electrovalves from the dry air tank and to the inlet (Figure 2). The humidity dependence
calibration mode produces a scale of increasing humidity steps in the evaporation chambers (e.g. from 100
ppmv to 1000 ppmv, through steps of 100 ppmv for 50 minutes for each standard). The details of the
sequence (standard type, humidity level and duration of each step) is defined in a text file by the operator
from the Raspberry interface, the Raspberry being itself connected to Ethernet for remote access.

The Picarro L2130-i analyser has an External Valve Sequencer, which is able to turn on/off up to six

electrovalves and create loop sequences with defined durations for each step of the sequence. This tool
can be diverted from its original purpose by using it as a 6-digit code: each of the humidity generator state
is associated with a code. When the Picarro Valve Sequencer matches one of the state code, this state is
triggered on the humidity generator. This eases both the operator's activities and the data post-treatment,
because the current valve status - thus the calibration instrument state - is saved in the analyzer output
data file, in the "ValveMask" column. The Raspberry inside the LHLG reads the Valve Sequencer state code
using the Picarro's Remote Control Interface (a RS232 serial connection through one of the rear-face DB9
connector).

| States (min) | Flow FCA (sccm) | Flow FCB (sccm) | Valve A | Valve B | Syringe Pump (μL/min) | Inlet Valve | Dry air Valve | Pressure controller (mbar) | Pressure Pump for exhaust | Double 3-way valve |
|---|---|---|---|---|---|---|---|---|---|---|
| Outside air (1100) | 0 | 0 | Closed | Closed | 0 | Open | Closed | Off | Off | To chamber |
| Drying (20) | 400 | 400 | Open | Open | 0 | Closed | Open | Off | Off | To chamber |
| Boost (0.7) | 300 | 300 | Open | Open | Infuse at 2.5 | Closed | Open | 905 | On | To chamber |
| Standard A (50) | 300 | 150 | Open | Closed | Infuse at 0.25 | Closed | Open | 905 | On | To chamber |

| Standard B (50) | 150 | 300 | Closed | Open | Infuse at 0.25 | Closed | Open | 905 | On | To chamber |
|---|---|---|---|---|---|---|---|---|---|---|
| Reset (1) | Closed | Closed | Closed | Closed | Withdraw max speed | Open | Closed | Off | Off | From standard |

**Table 1**: *Typical routine sequence of measurements + calibration for two standards A and B at 1000 ppmv for a measurement site located at sea level. No mixing occurs between standards A and B during steps "Standard A" and "Standard B" (see supplementary text S1).*

*Note that the humidity dependence mode and the expert mode can also be included in the valve sequencer but are not used in a daily calibration routine.*

A set of tools has been developed to quickly check daily calibration. In the field, analyzer and LHLG data are archived daily and sent to the laboratory, i.e. at LSCE, Gif sur Yvette. They are checked semi-automatically once a week to warn maintenance personnel in the event of a malfunction.

## 3-    Performance of the instrument

The stability of the instrument has been tested over a large range of parameters. We show an example in Table 2. We modified the air flow associated with standard A (the same results can be obtained with standard B) between 200 and 400 sccm with an air flow on channel B of half the value of channel A. The infusion rate was varied between 0.03 and 0.14 μL/min in order to produce humidity levels of 400 and 800 ppmv. The 1σ standard deviations observed over 10 minutes plateaus are comparable to the standard deviation obtained when the air flow is set to 300 sccm.

| Air flow (sccm) | Infusion rate (μL/min) | Humidity (ppmv) | 10 minutes 1σ standard deviation for humidity (ppmv) | $\delta^{18}O$ (‰) | 10 minutes 1σ standard deviation for $\delta^{18}O$ (‰) |
|---|---|---|---|---|---|
| 200 | 0.07 | 808 | 1 | -7.88 | 0.89 |
| 300 | 0.11 | 851 | 2 | -7.73 | 0.85 |
| 400 | 0.14 | 818 | 2 | -7.95 | 0.90 |
| 200 | 0.03 | 374 | 1 | -8.45 | 1.92 |
| 300 | 0.05 | 411 | 2 | -9.16 | 1.64 |
| 400 | 0.07 | 415 | 3 | -9.05 | 1.59 |

**Table 2** : *Evolution and stability of humidity and $\delta^{18}O$ (same water used for the different tests) for different syringe infusion rates and dry air flows.*

For routine measurement, air flow and infusion rate have been adjusted to optimize the stability of the
generated vapor while minimizing the dry air consumption. The LHLG is thus able to generate stable levels
of humidity (drift lower than 20 ppmv over one hour and $1\sigma$ below 10 ppmv over 10 minutes) from 70
ppmv to 2,400 ppmv following the optimal set-points shown in Table 3.

| Humidity (ppmv) | Infusion rate (µL/min) | Dry Air flow (sccm) |
|---|---|---|
| 80 | 0.01 | 300 |
| 160 | 0.02 | 300 |
| 320 | 0.04 | 300 |
| 800 | 0.1 | 300 |
| 1200 | 0.15 | 300 |
| 1600 | 0.2 | 300 |
| 2400 | 0.3 | 300 |


**Table 3:** *Set-points for water infusion rate and dry air flow at a temperature of 20°C.*

**3-1- No fractionation during water vaporization in the cavity**
We have checked that there was no fractionation of the water during its transfer from the bottles to the
syringe pump, then from the syringe to the moist air generated in the vaporization chamber through the
following tests.
First, the isotopic composition of three different lab-standards calibrated against VSMOW at LSCE ($H_2O$-
$CO_2$ equilibration followed by IRMS for $\delta^{18}O$; Cavity RingDown Spectroscopy for $\delta D$; calibrated every 3
years using VSMOW and VSLAP provided by IAEA) have been compared, after their generation by the
present LHLG and by the commercial SDM, both at a humidity of 2,000 ppmv over 50-min time spans. The
measured $\delta^{18}O$ and $\delta D$ values agreed to within 0.5‰ and 2‰, respectively, for the 3 lab-standard waters
calibrated against VSMOW: EPB ($\delta^{18}O$ = -6.24 ‰; $\delta D$ = -43.6 ‰), NEEM ($\delta^{18}O$ = -33.50 ‰; $\delta D$ = -257.2 ‰),
FP5 ($\delta^{18}O$ = -48.33 ‰; $\delta D$ = -383.5 ‰). Second, the measured isotopic composition of the same standard
(FP5) generated at different humidity levels between 1,000 and 2,400 ppmv by the SDM and the LHLG
show the same $\delta^{18}O$ ($\delta D$) evolution with humidity within respective uncertainties (Supplementary Figure
S1).

**3-2- Stability of the water vapor delivery and associated water isotopic composition**

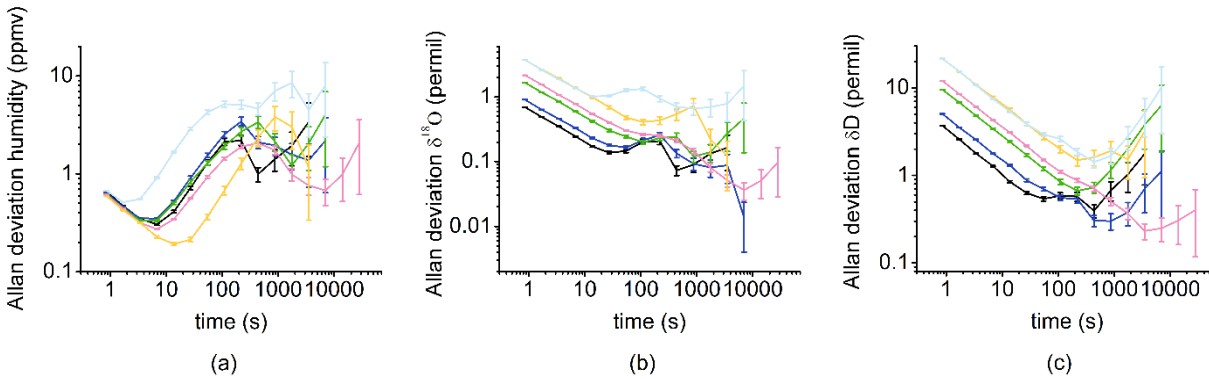

(a)  (b)  (c)

**_Figure 3:_** _Allan deviation over 4 hours for different humidity levels (black 1,080 ppmv; dark blue 770 ppmv; green 400 ppmv; pink 320 ppmv; yellow and light blue 170 ppmv) for humidity (a), $\delta^{18}O$ (b) and $\delta D$ (c)._

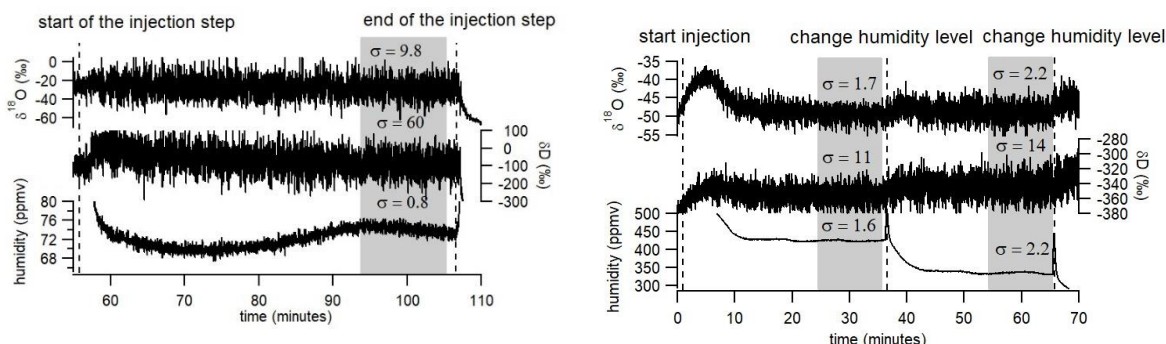

**_Figure 4_**: _Records of $\delta^{18}O$, $\delta D$ and humidity over 3 humidity plateaus (72 ppmv on the left, 425 and 335 ppmv on the right) obtained with the LHLG. The grey rectangles indicate the period (10 min) over which the average values are kept for calibrating the data generated by a L2130-i analyzer._

A proper approach to quantify the stability of our system is to use the Allan variance defined as:

$$\sigma_y^2(t) = \frac{1}{2}(\langle y_{n+1} - y_n\rangle^2) \qquad\qquad \text{(eq. 3)}$$

where $y_n$ are the successive measurements over a period t.

An Allan variance plot as a function of averaging time is indeed useful to determine the optimal time over which the sample humidity and the isotopic composition should be averaged to obtain a precise determination (low standard deviation) and avoid drift. Figure 3 displays the Allan deviation (square root of the Allan variance) in $\delta^{18}O$, $\delta D$ and humidity obtained by running a long plateau of standard A or standard B in the "infuse" mode over 4 hours for different humidity levels. The humidity variance always stays below 10 ppmv over the 4 hours test and the $\delta^{18}O$ and $\delta D$ Allan deviations display minimum values below 1 ‰ and 7 ‰ respectively. The minimum value for the $\delta^{18}O$ and $\delta D$ Allan deviation is generally

obtained for about 15 minutes of measurement. While the Allan deviation of $\delta^{18}O$ and $\delta D$ is dependent on
the analyzer used, we observe that the Allan deviation at 1000 s (17 minutes) for $\delta^{18}O$ and $\delta D$ also depends
to some extent on the humidity level: the lowest levels are obtained for humidity levels of 770-1,080 ppmv
and the highest levels are obtained for humidity level of 170 ppmv.
In the routine mode (Figure 4), we perform plateaus of 30 to 50 minutes (50 minutes when the instrument
is unattended since the time to reach the plateau varies between a few minutes to 30 minutes). We then
select the last 10 minutes before the following switch of the instrument to measure the average level of
humidity and the isotopic ratios, $\delta^{18}O$ and $\delta D$. We also calculate the associated standard deviations and
reject the values if the humidity standard deviation exceeds 30 ppmv over these last 10 minutes. In Figure
4, one observes that the standard deviations for humidities generated in the routine mode are actually
much lower. The corresponding standard deviations for the isotopic ratios ($\delta^{18}O$ and $\delta D$, see values
indicated in Figure 4) increase with decreasing humidity, reflecting the decrease of the molecular
absorption signal recorded by the L2130-i laser analyzers. This has an obvious impact on the determination
of the relationship between humidity and water vapor isotopic composition.
The performance of the present LHLG can be compared to the performance of the SDM (see
Supplementary Figures S1 and S2). First (Figure S2), a comparison has been performed at a humidity level
of 800 ppmv, for which we have numerous daily calibrations performed with a SDM from a 4.5 years field
deployment in Svalbard (Leroy-Dos Santos et al., 2020). The best SDM performance displays a standard
deviation $1\sigma$ of 31 ppmv, which is significantly worse than the performance of the LHLG (standard
deviation $1\sigma$ lower than 10 ppmv on average and down to 2 ppmv for 30% of the generated humidity
plateaus). Second (Figure S1), while we measure the same influence of humidity on measured $\delta^{18}O$ and $\delta D$
either with the SDM or with the LHLG, the $1\sigma$ values on humidity levels are much larger for the SDM than
for the LHLG.

**3-3-  Determination of the influence of humidity on water vapor isotopic composition**

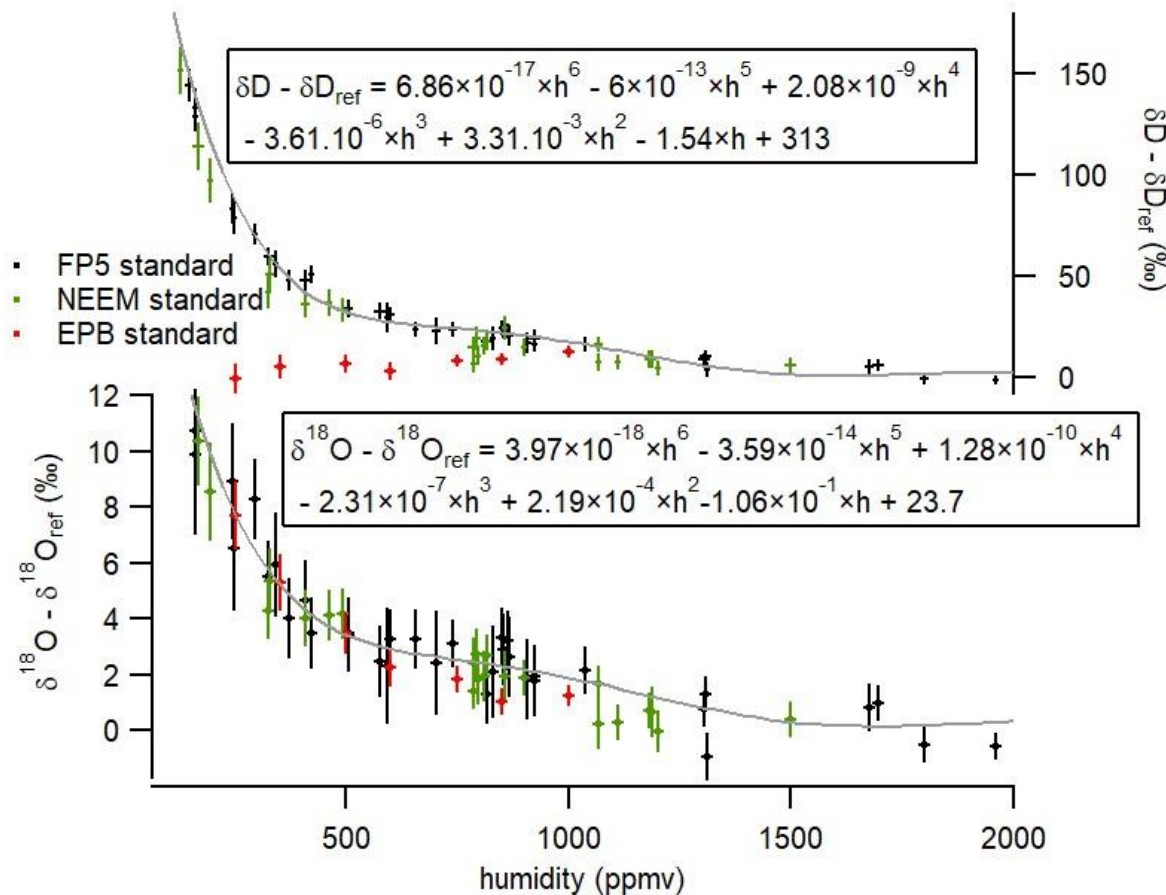


**_Figure 5_**: *Influence of humidity on the isotopic composition ($\delta^{18}O$ and $\delta D$) of the vapor obtained with the*

*LHLG with 3 water lab-standards. The error bars are calculated as the standard deviation ($1\sigma$) over the*
*generated values by the L2130-i instrument during 10 minutes at 1 second resolution (i.e. without any pre-*
*averaging of the raw dataseries). The $\delta^{18}O_{ref}$ and $\delta D_{ref}$ are the values of the injected water standards at*
*2,000 ppmv. The grey lines represent the polynomial fits for the influence of humidity on the water isotopic*
*composition (equations 4 and 5 also written on the graph).*

Contrary to the commercial SDM, which hardly produces stable and reproducible humidity levels below
500 ppmv, the LHLG was able to daily produce stable 10-minute humidity plateaus over the range from 70
ppmv to 2,400 ppmv with an associated standard deviation of the order of 10 ppmv over more than one
year at the Concordia and Dumont d'Urville stations (installation in December 2018). The stability of the
LHLG allows a robust quantification of the L2130-i analyzer drift thanks to a daily measurement of the
same water isotopic standard reference (see Table S2 showing actually no measurable drift over a 3-week
period). It also permits the characterization of the measurement non-linearities observed at low-humidity
(Figure 5). The more than one-year long Concordia and Dumont d'Urville datasets showed that the
humidity dependence of $\delta^{18}O$ and $\delta D$ did not vary measurably. The uncertainty of the obtained calibration
curve can be attributed entirely to the L2130-i $\delta^{18}$O and $\delta$D measurements. In other words, the uncertainty
bars in the horizontal (x-) axis in Figure 5, associated with the LHLG, are negligible.
Our data show a result already observed in Weng et al. (2020): while the dependency of $\delta^{18}$O and $\delta$D to
humidity is similar for low $\delta^{18}$O and $\delta$D lab-standards (NEEM and FP5), we observe a different behavior for
the $\delta$D vs humidity relationship for the high $\delta^{18}$O and $\delta$D lab-standard EPB. This result strengthens the
recommendation of Weng et al. (2020) to use two water standards in the range of the measured water
vapor isotopic composition to best calibrate our final data. In our case, our applications were in Antarctica,
so that we used our two lowest lab-standards (NEEM and FP5). For the two standards and for this particular
Picarro L2130-i (results are expected to depend on the instrument), the same dependency of isotopic
composition vs humidity is observed. We express this dependency as the relationship between the
difference in $\delta$D or $\delta^{18}$O between the measured value at the given humidity and the value of the same
standard measured at a humidity of 2,000 ppmv. The experimental data for NEEM and FP5 from Figure 5
are fitted through polynomial functions with respect to humidity h (in ppmv):

$\delta^{18}O - \delta^{18}O_{ref} = 3.97 \times 10^{-18} \times h^6 - 3.586315 \times 10^{-14} \times h^5 + 1.2843645994 \times 10^{-10} \times h^4 - 2.3087753445094 \times 10^{-7} \times h^3$
$+ 2.1857285350473100 \times 10^{-4} \times h^2 - 0.10603325432255400000 \times h + 23.7$  (eq. 4)

$\delta D - \delta D_{ref} = 6.859 \times 10^{-17} \times h^6 - 6.0047709 \times 10^{-13} \times h^5 + 2.0790331349 \times 10^{-9} \times h^4 - 3.61319302207374 \times 10^{-6} \times h^3$
$+ 3.30716141498371 \times 10^{-3} \times h^2 - 1.53651645114701 \times h + 313$ (eq. 5)

These curves are valid only for a given Picarro analyzer and for humidity higher than 70 ppmv and lower
than 2,000 ppmv. Outside this calibration range, the extrapolation of the polynomial function may lead to
anomalous corrections.

After this correction, the measured values corrected from humidity dependence are corrected using the
comparison of the measured values of the 2 standards at 2,000 ppmv to their VSMOW calibrated values
as explained in section 3.5 below.

**3.5- Accuracy of the system and calibration on the VSMOW-VSLAP scale**
The accuracy of the system has been addressed performing a 2-standard calibration and measuring a third
standard treated as an unknown. We used two lab-standards calibrated vs VSMOW on the VSMOW-VSLAP
scale with large $\delta^{18}$O and $\delta$D differences (EPB and FP5) and used the lab-standard NEEM, also
independently calibrated against VSMOW. The 3 lab-standards have been vaporized at 800 ppmv and
measured by the same L2130-i analyzer.

| Standard | VSMOW calibrated value | Measured value at 800 ppmv | Measured value corrected from |
|---|---|---|---|

|  |  |  | humidity dependence (Equation 1) |
| --- | --- | --- | --- |
| EPB | -6.24 ‰ | -8.27 ‰ | -10.78 ‰ |
| NEEM | -33.5 ‰ | -34.48 ‰ | -36.99 ‰ |
| FP5 | -48.33 ‰ | -49.02 ‰ | -51.53 ‰ |


**Table 4**: *Comparison of measured vs VSMOW calibrated $\delta^{18}O$ values for 3 standards measured with a*
*Picarro analyzer after generation of water vapor using the LHLG.*

We used the measured and true values of EPB and FP5 to estimate the $\delta^{18}O$ value of the NEEM standard
from its measured value (Table 4). Using the linear relationship obtained from VSMOW calibrated EPB
and FP5 $\delta^{18}O$ vs measured EPB and FP5 $\delta^{18}O$ values (Figure S3) following the recommendations of the
National Institute of Standards and Technology (NIST, reference material 8535a) leads to an estimated
NEEM $\delta^{18}O$ of -33.31 ‰ to be compared to the independently VSMOW calibrated value of -33.5 ‰.
Given the uncertainty of about 0.8-1 ‰ when measuring $\delta^{18}O$ around 800 ppmv, we can conclude that
the system is accurate.

**4-    Application**
The main application of this device is the interpretation of water isotopic profiles at dry sites, in particular
in polar regions. As shown in Figure 5, the influence of humidity on the measurement of the water vapor
isotopic composition with the L2130-i analyzer is large when humidity is below 1,000 ppm and increases
when humidity decreases. Even though the precise isotope ratio-humidity calibration curve is likely to be
different from one analyzer to another, all laser-based water isotope analyzers investigated to date have
shown a strongly non-linear response at low-humidity levels (Guilpart et al., 2017; Leroy Dos-Santos, 2020;
Weng et al., 2020). At the Concordia station, even in summer, humidity is generally below 1,000 ppmv
(Figure 6) so that the interpretation of the diurnal variability of the water vapor isotopic composition is
strongly affected by the dependency of the measured $\delta^{18}O$ and $\delta D$ signals on humidity. Figure 6 displays
such diurnal variabilities during austral summer 2018-2019 at Concordia and the consequently large
correction of the isotopic records (uncorrected in grey and corrected in black).

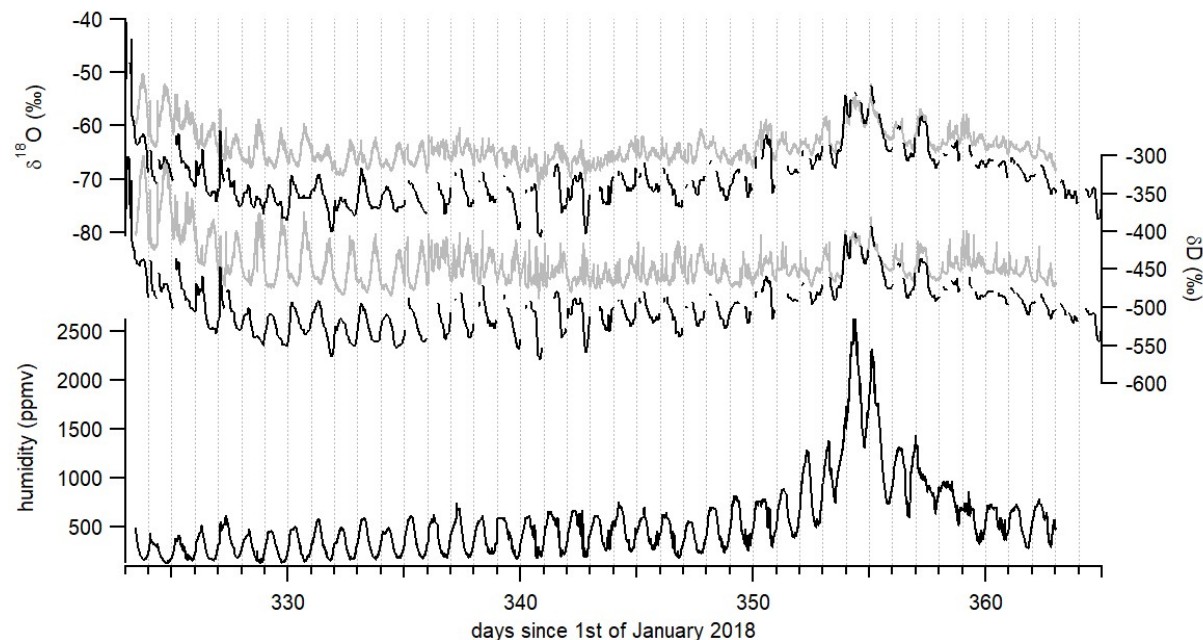


***Figure 6***: *$\delta^{18}O$, $\delta D$ and humidity records over December 2018 and beginning of January 2019. Raw isotopic*
*values are in grey. Corrected isotopic values at hourly resolution are in black after correction of the*
*influence of humidity on the water isotopic ratios and adjustment of $\delta^{18}O$ and $\delta D$ values on the VSMOW-*
*VSLAP scale using relationships between measured lab-standard values and known VSMOW calibrated lab-*
*standard values.*

The data clearly demonstrate the importance of the humidity correction which shifts the curves generally
to lower isotopic ratio values. However, the difference between uncorrected and corrected data is
particularly important in the observation of the diurnal variability, illustrated even better when zooming
in on a section of the data, as in Figure 7. When looking in detail at the diurnal variability in the raw $\delta^{18}O$
and $\delta D$ isotope data, some periods stand out with two identified daily peaks, one in phase with the
humidity peak (marked in red in Figure 7) and one occurring during the period of minimum humidity
(marked in blue in Figure 7). The strong non-linearity of the calibration curve of Figure 5 suggests that
artificial peaks in $\delta^{18}O$ and $\delta D$ could be due to changing humidity levels. Indeed, after correcting the data
for the humidity dependence of the analyzer (black curve in Figure 7), the isotopic peaks occurring during
humidity minima are diminished or disappear altogether, while the peaks occurring during humidity
maxima are amplified. More strikingly, the phase of the signal changes by practically 180° over some
periods. Whereas the raw isotope signal peaks during the night, the corrected record shows higher isotope
ratios during daytime. The diurnal variability recorded on both raw and corrected isotopic values during a
period with higher humidity level, hence when the isotope ratio-humidity correction is smaller (around
day 355 in figure 6), also shows that the $\delta^{18}O$ ($\delta D$) diurnal cycles are indeed in-phase with the humidity
cycle. This result confirms the correlation between humidity cycles and $\delta^{18}O$ and $\delta D$ of the water vapor at
the daily scale at Concordia as reported by Casado et al. (2016). We thus conclude that the anticorrelation
observed between $\delta^{18}O$ ($\delta D$) and humidity in the raw data (highlighted in blue in Figure 7) during periods
of low-humidity is an artefact due to the influence of the humidity level on the vapor isotopic
measurements by the L2130-i analyzer.

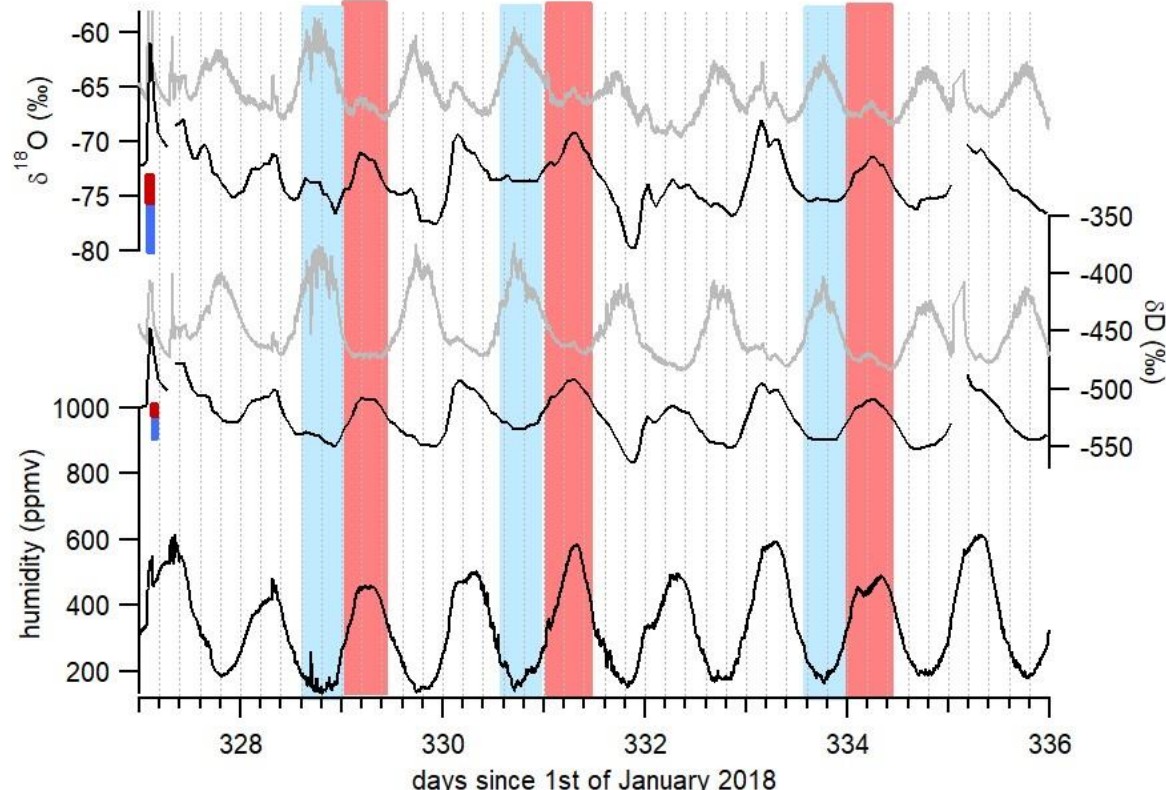


**_Figure 7_**: _Focus on diurnal variability of $\delta^{18}O$, $\delta D$ and humidity recorded at Concordia. Grey curves show the_
_raw measurements and black curves the corrected records. The red (blue) bars indicate the calculated_
_uncertainty due to the isotopic ratio vs humidity dependence (Figure 5) on the corrected $\delta^{18}O$ and $\delta D$ values_
_during periods with maximum (minimum) humidity. The red (blue) rectangles indicate half day with_
_maximum (minimum) humidity._


**5-     Conclusion**

We have developed an autonomous instrument for low-humidity generation (70 to 2,400 ppmv) with
controlled water vapor isotopic composition specifically aimed at carrying out continuous measurements
of the water vapor isotopic composition using a laser-based spectrometer in regions characterized by very
low-humidity, such as polar regions. If needed, an interface permits to conveniently connect the new LHLG
to the valve sequencer port of commercial Picarro instruments. After more than one year of routine
operation on two Antarctic sites (Dumont d'Urville and Concordia), this instrument has proven to be very
reliable and robust. It consistently generates stable humidity levels with a 1$\sigma$ variability lower than 10
ppmv over more than 10 minutes. Besides, its performance is significantly better than that of the Picarro
SDM at low humidity.
We used this instrument for the calibration of our water isotopic data with a special focus on accurately
quantifying the influence of humidity on the measured isotopic composition of the water vapor. This effect
is huge at low-humidity. We showed that this has an important impact on the interpretation of the diurnal
cycles of $\delta^{18}O$ and $\delta D$ in the water vapor at the Concordia station at humidity below 1,500 ppmv. We were
able to confirm that, at this site, the diurnal $\delta^{18}O$ and $\delta D$ variability is actually correlated with humidity
variability, which would not have been possible without the new LHLG instrument.
Finally, the development of such an instrument is an important step forward to a better understanding of
the transfer function between climate parameters and the isotopic composition of deep ice cores from
the remote East Antarctic plateau, especially in the context of the new program "Beyond EPICA". It should
be completed by ongoing development of laser spectrometers better adapted to low-humidity levels, such
as those based on the technique of Optical Feedback Cavity Enhanced Absorption Spectroscopy (OFCEAS)
(Casado et al., 2016; Landsberg, 2014; Landsberg et al., 2014).

**Code availability**
The software (HumGen) can be downloaded on line (https://github.com/ojsd/humgen;
https://doi.org/10.5281/zenodo.4003465).

**Competing interests**
The authors declare that they do not have any competing interest.

**Author contributions**
CLDS, MC, FP and EK designed and built the instrument. OJ realized the software interface development.
CLDS, MC and AL installed the instrument in Antarctica and tested it extensively. EK, SK, MF, AL and EF
tested the instrument in the laboratory. AL wrote the manuscript with the help of all co-authors.

**Acknowledgments**
The development presented in this manuscript is largely inspired from the initial PhD work of Janek
Landsberg which we gratefully acknowledge here. The research leading to these results has received
funding from the Antarctic Snow program of the Fondation Prince Albert II de Monaco, the ANR EAIIST and
CNRS-LEFE program ADELISE. The deployment of this instrument in the field was made possible through
the logistic support of the NIVO2 & ADELISE IPEV programs. We thank the two reviewers for their useful
comments which greatly improved the manuscript.

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
