# Peer review of "A dedicated robust instrument for water vapor generation at low-humidity for use with a laser water"

_Atmospheric Measurement Techniques, 2020_

## Referee Comment (RC1) · Anonymous Referee #1 · 17 Nov 2020

General comment The manuscript describes a custom evaporator, specifically designed for low humidity areas. Such device was tested in Antarctica for more than one year. The main difference with published literature that report similar devices (e.g. Gkinis et al., 2010 for CFA system) is that water flux into the evaporating chamber is injected with a syringe (which allows very precise tuning of mass flow). Secondly, there is a very precise control of dry gas flux and pressure inside the evaporation chamber. The study presents one of the few instruments (with the micro drop device of Iannone et al. 2009) that can provide stable vapor signal at very low humidity values, suitable

for isotopic analysis with laser analyzers. For this reason I think that the manuscript is well suited for AMT and I only suggest minor revision.

Technical comments: 1- This study represents an extension of the work shown in Landsberg (2014). However, in his study a strong influence of the lead screw rotation was identified as one of the main source of noise in the water vapor signal at low humidity. Is the choice of the pump a critical point for the development of the instrument? There are other critical points for designing/replicating this device that must be taken into account (e.g. how important is the choice mass flow and pressure controller? The authors should discuss this point e.g. in section 2.2.

2- Since large stability of water vapor flux is expected by such device, one would expect an analysis of the stability of water vapor signal, at least in terms of the mixing ratio. An analysis of stability could be the identification/absence of trend during different humidity steps or the analysis of mixing ratio standard deviation as a function of different instrumental configuration (e.g. dry air flux, syringe speed etc). A stability analysis would provide additional evidence of the robustness/reliability of the instrument.

Other minor comments

L136 Period.

L147-148 What is the reason for using fractionation factors of Cappa et al. (2003)?

Figure 2. Check part labels (A/B) because different names are used in text, in table 1 (F1, G1 etc..) and in Figure 3.

L245 In relationship with my technical comment #2: what stable means? No trend observed in mixing ratio? Low RSD?

Figure 4. It is not clear how long it takes the signal to stabilize and whether this stabilization period is related to the humidity level. From Fig.4 it seems so, because injection steps are characterised by different lengths.

Section 3.2 Maybe, two words here (or a simple scatter plot, boxplot) about a possible relationship between precision and humidity level could help the reader understand the stability of the system.

Section 3.3, Figure 5 and Section 4: humidity correction is important. I would like to point out to the authors that a correction based on the difference between observed and reference values for a single standard might not be enough, as recently highlighted out in Weng et al. 2020. Under the perspective of the influence of both humidity and isotopic composition, the correction function should represent a plane and not a line (as one could expect from Figure 4). Moreover, from Figure 4 it is not clear if the scatterplots reports the difference between a single standard (obs-ref) or for different standards (with different isotopic composition) because the plural "standards" word was used. Maybe the authors could highlight that under low humidity and low isotopic variability, the humidity response of the instrument can be determined by analysing one (or two) standard(s).

Cappa, Christopher D., et al. "Isotopic fractionation of water during evaporation." Journal of Geophysical Research: Atmospheres 108.D16 (2003).

Gkinis, Vasileios, et al. "A continuous stream flash evaporator for the calibration of an IR cavity ring-down spectrometer for the isotopic analysis of water." Isotopes in environmental and health studies 46.4 (2010): 463-475.

Iannone, Rosario Q., et al. "A microdrop generator for the calibration of a water vapor isotope ratio spectrometer." Journal of Atmospheric and Oceanic Technology 26.7 (2009): 1275-1288.

Landsberg, Janek. "Development of an OF-CEAS laser spectrometer for water vapor isotope measurements at low water concentrations." University of Groningen (2014).

Weng, Yongbiao, Alexandra Touzeau, and Harald Sodemann. "Correcting the impact of the isotope composition on the mixing ratio dependency of water vapour isotope

measurements with cavity ring-down spectrometers." Atmospheric Measurement Techniques 13.6 (2020): 3167-3190.

---

## Referee Comment (RC2) · Anonymous Referee #2 · 6 Dec 2020

**1 Overview**

The manuscript "A dedicated robust instrument for water vapour generation at low humidity for use with a laser water isotope analyser in cold and dry polar regions." by Christophe Leroy-Dos Santos et al., deals with a very non-trivial challenge in the area of in situ isotopic analysis of water vapour in polar regions using Infra Red Spectroscopy. The very low water concentration levels, typical for the atmospheric air in polar areas and in particular the very dry regions of East Antarctica pose a big challenge

for the optical spectrometers used for in situ measurements of the isotopic composition of the vapour. One part of the challenge has to do with the very low signal to noise ratio in the measurement that results in a poor analytical precision. The second challenge relates to the observed dependence of the isotopic measurement on the water concentration level of the sample in the optical cavity of the spectrometer. In the more modern versions of these Cavity Ring Down Spectrometers, this dependence is addressed with in-factory corrections that typically take into account changes in the absorption spectrum and can provide high quality measurements down to  $[H_2O]\approx 2000$  ppmv. Below this level, additional corrections need to be performed usually requiring the generation of a vapour stream with stable values for  $[H_2O]$  and  $\delta^{18}O$  and  $\delta D$ .

In this work, Santos et al propose a water vapor generator that is able to provide a stream of vapor with stable  $[H_2O]$  and isotopic composition, using a combination of syringe pumps and an evaporation chamber aided by flow control instruments. This is a paper that belongs to AMT, providing a tool that is important for polar research. While the scientific significance of the work is undoubted, the manuscript lacks clarity to the extend that the scientific quality, even if apparent, is not easy to judge. Additionally, the presentation quality is fair and the manuscript lacks the use of standard nomenclature commonly seen in technical publications of this type in AMT and other instrumentation oriented journals.

Below I comment on some of the major issues I believe the authors need to address and I propose some changes and experiments that in my view are essential for this study. Considering the importance of the problem the manuscript deals with, I would like to see it eventually published in AMT, after the authors proceed with some major revisions. Perhaps it may in fact be easier and more clean if they start fresh with a new submission, though this is something the editor and the authors should decide.
**2 Remarks**

**2.1 Evaporation Model**

The theoretical foundation of the vapor generator presented here is described in section 2.1 of the manuscript. It is based on the assumption that a constant flow of water ending on a syringe needle tip combined with a finely controlled flux of dry air will produce an isotopically stable stream of vapor via a zero fractionation process. A prerequisite for this, is that the size of the droplet remains stable throughout the experiment. It is still a question to me how a droplet that sustains its size by constantly loosing mass on its surface (regained by the incoming liquid water flow) via evaporation yields a vapor stream that has the same isotopic composition as the liquid. This is a typical Rayleigh evaporation likely with a strong kinetic component due to the very low humidity of the carrier gas and the quick -but incomplete- removal of water molecules from the droplet. Even though the treatment of the model has been presented in previous works, this is so central to this system that at least an appendix with more information is needed.

- 2.1.1 Some points to consider
  - How is the size of the droplet controlled when there is no camera or some other monitoring tool inside the evaporation chamber?
  - The manuscript mentions in Line147 that the fractionation factor by Cappa et al (2003) are used. But why if there is no fractionation?
  - The manuscript claims that under steady-state the isotopic composition of the generated vapour does not depend on the injected flux of water nor the specific humidity. Since this is a technical paper I would appreciate a simple experiment where the flux of the carrier gas is ramped up and down allowed to reach steady
state. I have a very hard time seeing how this experiment will produce a vapor stream of the same isotopic composition equal to the  $\delta^{18}$ O of the liquid water. Similarly if the specific humidity of the carrier gas is changed one ends up with a simple mixing experiment (Mook , 2000) where the resulting vapour isotopic composition naturally will be different.

- Throughout the whole manuscript, I have not seen a note on the temperature of the evaporation. This is a critical parameter affecting the efficiency of the evaporation, the saturation vapour pressure of the mixture and the fractionation factors in the (Cappa et al , 2003) parameterization. Since there is no active control and regulation of the temperature, all these parameters will vary.
- 2.2 Presentation-Standard nomenclature

Any physicist or engineer with moderate training in laboratory instrumentation should be able to look into the flow diagram of a manuscript in AMT or any other equivalent journal and get a basic idea of the method described in it. It is a very important element of a publication of this type, therefore it is my view that the authors should use standard P&ID nomenclature. The current flow diagram is a collection of coloured boxes from which little can be understood. Part of the text uses the photograph supplied (Figure 3) to explain the system something that confuses the reader even more.

**2.2.1 Points to consider**

• There are standard P&I symbols for valves, pumps, gauges etc that should be used, while colour should be added only if it aids in better explaining the system
and not cosmetically. Standard names for the components should be used. For example three pumps with three different descriptions (pressure pump, pump and picarro pump) are found in the block diagram when they (most likely) are of the very same type ie vaccum pump.

- One of the main elements of the paper according to the authors is the "double 3way valve" added to the system. The valve's type is a 6-port switching valve and a standard symbol for it exists that should be used in this manuscript. It is central to describing the cycles of the system. Feel free to use the supplied (arguably not perfect) Fig. 1 that I am including and describes the valve operation for the syringe filling position.
- Flows and pressure set points for the regulators also belong to the P&I diagram and since the authors claim that they are important to the operation of the system should be given. A table with the ID numbers of the control units and their set points would be very much appreciated. Currently Table 1 blends names of components from Figure 2 and Figure 3 so the reader has to guess. This is not informative.

**2.3 Experimental System–Explanation of operation**

Section 2.2 of the manuscript contains information on the principle of operation. One of the puzzling pieces of information in this section is the carrier gas flow. Following the block diagram and the information in Table 1, we see that for the "Drying Mode", the Flow Controllers A and B sustain each a 400 sccm-1 flow through the system. The block diagram does not indicate any open splits and for this particular mode the inlet valve is closed and the exhaust vacuum pump is disabled. This is an immense flow for the standard of a picarro spectrometer that normally can sustain its cavity pressure with sample flows in the order of 30-100 sccm-1.
Similarly, during the injection of Standard A or Standard B, the flow is in the order of 450 sccm-1, a very high flow level. In this mode though the exhaust vacuum pump is enabled, however we see that the lines of Standard A and Standard B are connected. Thus in Standard A mode, valve B is closed and the generated vapour from standard B, hopefully is evacuated via the exhaust vacuum pump. Is the pressure of 905 mbar enough to make sure that all the 150 sccm-1 of standard B are off the system and not mixed with the vapour from Standard A? Why was this value chosen and what kind of tests assure that the two lines are never mixed even though physically connected. The safe way to proceed here would be to simply isolate the two channels, remove the pressure regulator and the vacuum pump and simply install two 3-way valves in the position of valves A and B (which I assume are On/Off???) on which one port goes to the instrument inlet and the other is simply open to the atmosphere in an open split configuration.

- 2.3.1 Points to consider
  - A more thorough description of the flow path is needed and the issue of possible cross contmination between Standard A and B should be addressed.
  - The flows are very high How does the picarro cope with this condition?
  - There is no information on the volume of the evaporation chambers and no mention regarding the temperature of the system.
  - There are references to a two standard calibration protocol in this section. It is not exactly clear if this concerns some water concentration correction calibration or a linear slope SMOW-SLAP calibration—see specific comment on the notable absence of any reference to SMOW-SLAP below.
  - I assume that the system is also used in order to perform a SMOW-SLAP calibration. How are evaporation/fractionation effects in the standards' vials handled?
Based on Figure 3 we see no precautions concerning this.

**2.4 Evaluation–Measurement Stability and precision**

The manuscript lacks important information on the evaluation of the performance of the system. Section 3.1 assures the reader that no fractionation is observed during the generation of the vapour and its transfer in the flow lines while the agreement with the commercial standard delivery module is within 0.2 and 1‰ for  $\delta^{18}$ O and  $\delta$ D respectively. It is almost impossible to judge these discrepancies based on figure 5 and in fact when looking closely it is rather difficult to see how the differences between the two systems are of this order.

With respect to figure 4 and the evaluation of the precision for  $[H_2O]$  and  $\delta^{18}O/\delta D$  I strongly believe that the manuscript should include a proper Allan variance test (Werle et al , 2011; Steig et al , 2014). It is obvious that the system experiences drifts, whose origing is the vapour generator. Calculating the standard deviation on a 10 min window of a time series that obviously has a changing mean value looks and feels like possible cherry-picking.

**2.4.1 Points to consider–Suggestions**

- Show better evidence of zero fractionation. If you choose the SDM as a measure of comparison show a detailed comparison and plot the results clearly. The axes of fig 5 span 300 ‰ for  $\delta$ D. It is impossible to say anything.
- Calculate the Allan variance of the system for 4-5 [H2O] levels. Let one water run for several hours and so you get a more honest idea about precision and drifts for all three parameters under consideration. It is a very simple experiment that can provide a valuable insight and by looking in the bibliography it is a rather standard evalluation tool for laser spectroscopy based measurements.
**2.5 Water concentration correction**

The water concentration correction is described in section 3.3 with Figure 5 being the main source of information for this part of the analysis. I would consider this one of the most important sections in the paper and unfortunately it spans only one paragraph. It is repeated in the manuscript that the system presented here is superior to the commercial standard delivery module below the level of 500 ppm. How can we see this when the SDM measurements in this are of  $[H_2O]$  are not presented (measurements stop at  $[H_2O] \approx 2000$  ppm)?

The dataset in Figure 5 presents the difference of the raw isotopic value and a value that the authors call "reference" and "real". These words commonly refer to very specific things in isotope geochemistry and my guess is that the reference value is a SMOW-SLAP calibrated value (?). The term SMOW-SLAP calibration is not to be found at all in the manuscript. It is to some extent problematic that a water concentration correction measurement compares a raw value at a number of  $[H_2O]$  levels with a value post SMOW-SLAP calibration. Please see my comments on SMOW-SLAP in the next section.

The type of correction is not described in the manuscript. This is what this work is all about. There can be various approaches on how it can be done (see for example (Schmidt et al , 2010; Gkinis et al , 2010; Aemisegger et al , 2012)) but the authors need to be open and specific about what they did exactly. Moreover (Schmidt et al , 2010) suggest that the response to changing  $[H_2O]$  levels varies with the isotopic composition. I do not know how important this effect is for this present work, but I believe that the authors should perform two separate experiments with two different waters (technically the system offers this possibility). It would also strengthen their claims for a vary stable system if they show the full experiment with its raw data as a function of time and not only the averages.

There is also a claim in this section that the stability of the system allows via the
daily measurement of standards waters to quantify instrumental drifts. There is no strict evaluation of the stability of the system so far and I think that a proper Allan variance test at various levels of  $[H_2O]$  is necessary. But even in the absence of an Allan variance test the data given in Table 2 do not necessarily indicate instrumental drifts. All the isotope data in the table are within  $1 - \sigma$  of the noise level for a 10-min period mentioned in the caption and there does not seem to be a one way trend. So how is it possible to say anything about instrumental drifts? Also the noise levels mentioned in the caption of Table 2 are very different for roughly the same  $[H_2O]$  when one looks at the data of Figure 4 (4.5 versus 14).

**2.5.1 Points to consider-Suggestions**

- A clear description of the correction is missing. Some math is needed here.
- An experiment at two different isotopic levels showing the raw data versus time will show if there is an isotope effect in the water concentration dependence.
- A proper quantification of instrumental drifts (and this concerns the system as a whole and not only the spectrometer) can be done with a proper Allan variance test.
- 2.6 Lack of reference to SMOW-SLAP calibration

The main goal of building a water vapour generator as a peripheral for isotope measurements of water vapour is to be able to calibrate the dataset on the SMOW-SLAP scale. This is the only way to communicate and compare the measurements with other existing data sets and produce some science out of them. It is also even more important if the deuterium excess parameter will be studied as it is very sensitive to this calibration procedure. Therefore it appears very awkward that a manuscript dealing
with this topic does not include a single note, comment or reference to this very important step Of the measurement process. The dataset presented later in section 4 of the manuscript are impossible to evaluate if they are not calibrated in the SMOW-SLAP scale.

One more purpose of performing such calibrations, is that they can reveal possible accuracy issues in the instrumentation system. Given two standard waters one should be able to produce a calibration line and thereafter measure a third water of known isotopic composition treating it as an unknown. If the resulting value lies beyond the  $3 - \sigma$  range then there is likely something wrong with the system. That could be any part from the water standard storage to the water vapour generation system or the spectrometer itself. Currently there is no way to say anything about the accuracy of the system. With this in mind, section 4 of the manuscript is of very little use as the dataset is reported on some local instrument scale.

A SMOW-SLAP calibration experiment at various  $[H_2O]$  levels using the SDM and the current system would provide a proper comparison between the two systems and therefore it would be a very important addition to the manuscript.

**2.6.1 Points to consider**

- A proper treatment of the SMOW-SLAP calibration step is notably missing.
- Performing 2-standard calibrations and measuring a third water standard treated as an unknown will be a valuable -almost essential- addition to the manuscript, offering important information on the accuracy of the system.
- Since a lot has been written about the performance of the commercial SDM it will be proper to perform 2-standard calibrations for various [H2O] levels and compare the results.
I encourage the authors to revisit the manuscript and take the necessary steps as their contribution with this experimental system could be important for the community. The manuscript is to some extent rushed and some important aspects of the evaluation of the system are either missing or non-standard practices are followed. Some of the points I am raising here, are not only personal opinions but also guidelines of the International Atomic Energy Agency. I am confident that the authors can perform the necessary measurements/experiments and revise the manuscript.

**References**

- F. Aemisegger, P. Sturm, P. Graf, H. Sodemann, S. Pfahl, A. Knohl, and H. Wernli. Measuring variations of  $\delta^{18}$ O and  $\delta^{2}$ H in atmospheric water vapour using two commercial laserbased spectrometers: an instrument characterisation study. *Atmospheric Measurement Techniques*, 5(7):1491–1511–1491–1511, 2012.
- C. D. Cappa, M. B. Hendricks, D. J. DePaolo, and R. C. Cohen. Isotopic fractionation of water during evaporation. *Journal Of Geophysical Research-Atmospheres*, 108(D16):4525–4525, 2003.
- V. Gkinis, T. J. Popp, S. J. Johnsen, and T. Blunier. A continuous stream flash evaporator for the calibration of an IR cavity ring-down spectrometer for the isotopic analysis of water. *Isotopes In Environmental and Health Studies*, 46(4):463–475, 2010.
- W. Mook and W. Mook. Environmental Isotopes in the Hydrological Cycle: Principles and Applications, vol. I, IAEA. Unesco and IAEA, 2000.
- M. Schmidt, K. Maseyk, C. Lett, P. Biron, P. Richard, T. Bariac, and U. Seibt. Concentration effects on laser-based del(18)o and del(2)h measurements and implications for the calibration of vapour measurements with liquid standards. *Rapid Communications In Mass Spectrometry*, 24(24):3553–3561–3553–3561, 2010.
- E. J. Steig, V. Gkinis, A. J. Schauer, S. W. Schoenemann, K. Samek, J. Hoffnagle, K. J. Dennis, and S. M. Tan. Calibrated high-precision 17o-excess measurements using cavity ring-down
spectroscopy with laser-current-tuned cavity resonance. *Atmos. Meas. Tech.*, 7(8):2421–2435–2421–2435, 2014.

P. Werle. Accuracy and precision of laser spectrometers for trace gas sensing in the presence of optical fringes and atmospheric turbulence. *Applied Physics B-lasers and Optics*, 102(2):313–329–313–329, 2011.

**Supplement:**

[Figure]

Std1     Syringe1

ChamberP1

Std2

Syringe2

ChamberP2

Position 1
filling syringes

---

## Author Comment (AC1) · 2 Jan 2021

We thank the reviewers for very useful comments. We provide below the answers to the different comments and will revise the manuscript accordingly. Note that part of the answers as well as the need for a more detailed explanation of the theory associated with the vaporization of the drop is provided in a companion paper which is now available online (Kerstel, 2020). We will also add one co-author, Morgane Farradèche, who realized numerous of the additional tests requested by the reviewers.

**Reviewer 1**

Technical comments: 1- This study represents an extension of the work shown in Landsberg (2014). However, in his study a strong influence of the lead screw rotation was identified as one of the main source of noise in the water vapor signal at low humidity. Is the choice of the pump a critical point for the development of the instrument? There are other critical points for designing/replicating this device that must be taken into account (e.g. how important is the choice mass flow and pressure controller? The authors should discuss this point e.g. in section 2.2.

Indeed the choice of pump is important. First of all because it needs to provide an exceedingly small water flux, and secondly because it needs to do so with high stability. It is therefore noted that the oscillations that are in-phase with the lead-screw rotation were seen by Landsberg (2014) in an early prototype and were eliminated by better engineering, and in particular by the careful elimination of air-bubbles in the liquid water supply lines and syringe. This aspect is also addressed in our companion paper (Kerstel, 2020).

The precision of the pressure controller is not critical. Its purpose is to provide a steady inlet pressure to the optical spectrometer, also during syringe switching. Ideally the spectrometer is not sensitive to the inlet pressure.

The precision of the flow controller on the other hand does directly influence the precision of the volume mixing ratio (humidity level) produced by the instrument. Its precision ranges from about 0.5% at the highest flow setting, to almost 50% at the lowest air flow setting. In practice the device is used with an intermediate air flow of at which the precision of the controller is specified to be 1%. We observe, however, that in practice its short-term precision is much better. In any event, a precision of the order of 1% is normally comparable to the precision of the measurement of the humidity level with the optical spectrometer.

We will modify the text accordingly to provide quantitative information on these issues and cite the companion paper.

2- Since large stability of water vapor flux is expected by such device, one would expect an analysis of the stability of water vapor signal, at least in terms of the mixing ratio. An analysis of stability could be the identification/absence of trend during different humidity steps or the analysis of mixing ratio standard deviation as a function of different instrumental configuration (e.g. dry air flux, syringe speed etc). A stability analysis would provide additional evidence of the robustness/reliability of the instrument.

This issue was addressed through two different tests. First, we now present Allan variance obtained over 4 hours at different humidity levels by varying the syringe speed (Figure R1).

[Figure]

**Figure R1**: Allan variance over 4 hours for different humidity levels (blue 1080 ppmv; green 770 ppmv; black 400 ppmv; red 320 ppmv).

Second, we also test the stability of the system by varying the flow of dry air from 200 to 400 sccm (Table R1). The reported standard deviation (Table 1) is over a period of 10 minutes (last 10 minutes of the plateau) but the humidity and $\delta^{18}O$ values were stable over one hour (the experiment was stopped after one hour of stable humidity). We notice that the generated $\delta^{18}O$ values does not depend on the dry air flow nor on the infusion rate.

| Air flow (sccm) | Infusion rate (µL/min) | Humidity (ppmv) | stdev humidity over 10 minutes (ppmv) | $\delta^{18}O$ (‰) | stdev $\delta^{18}O$ over 10 minutes (‰) |
|---|---|---|---|---|---|
| 200 | 0.07 | 808 | 1 | -7.88 | 0.89 |
| 300 | 0.11 | 851 | 2 | -7.73 | 0.85 |
| 400 | 0.14 | 818 | 2 | -7.95 | 0.90 |
| 200 | 0.03 | 374 | 1 | -8.45 | 1.92 |
| 300 | 0.05 | 411 | 2 | -9.16 | 1.64 |
| 400 | 0.07 | 415 | 3 | -9.05 | 1.59 |

**Table R1** : Evolution and stability of humidity and $\delta^{18}O$ (same water used for the different tests) for different syringe infusion rates and dry air flows.

These results have been implemented in the revised manuscript.

Other minor comments

L136 Period.

L147-148 What is the reason for using fractionation factors of Cappa et al. (2003)?

In this paper we are interested in the steady-state behavior of the device. In this case, the isotopic composition of the incoming liquid water is by mass balance considerations equal to the isotopic composition of the evaporated water flux. The fractionation factor between the liquid and vapor phase is thus of no importance for the work presented here. The previously mentioned companion paper, however, deals with the dynamic behavior of the humidity generator and discusses the fractionation factors in detail. There we use estimations of the fractionation factors that are based on the studies by Cappa et al. (2003) and Luz et al. (2009). The work concludes that the effective fractionation factors are largely determined by their equilibrium counter parts, already determined with high precision by Horita and Wesolowski (1994), combined with the ratio of diffusivities and a parameter that describes the flow regime (from laminar to slightly turbulent). The diffusivities were determined by Merlivat (1978), Cappa et al. (2003) and more recently by Luz et al. (2009). We used the values by Cappa et al. (2003), but the difference is minimal for our purpose. Cappa et al. (2003) predicts only slightly lower values of the effective fractionation factor in the laminar limit compared to other estimates.

As the fractionation factors are not necessary for this study, we will remove the reference to Cappa in this study. The dynamical aspects are dealt in the companion paper.

Figure 2. Check part labels (A/B) because different names are used in text, in table 1 (F1, G1 etc..) and in Figure 3.

>> The scheme has been fully changed, see answer to reviewer 2 and associated Figure R3.

L245 In relationship with my technical comment #2: what stable means? No trend observed in mixing ratio? Low RSD?

>> Indeed, stable means both no trend in the mixing ratio (< 20 ppmv/hour) and low RSD (< 10 ppmv over 10 minutes). This will be explained in the revised manuscript.

Figure 4. It is not clear how long it takes the signal to stabilize and whether this stabilization period is related to the humidity level. From Fig.4 it seems so, because injection steps are characterised by different lengths.

>> Actually, it can be variable between a few minutes only to 30 minutes in worst cases. In best case, we have stabilization in a few minutes but it may take more time for undetermined reason. This is the reason why, in routine mode (i.e. when the user is not checking visually the stability of the plateau), we run 50-minute long plateau. This will be explained in the revised text.

Section 3.2 Maybe, two words here (or a simple scatter plot, boxplot) about a possible relationship between precision and humidity level could help the reader understand the stability of the system.

>> There is actually no relationship between precision and humidity level.

Section 3.3, Figure 5 and Section 4: humidity correction is important. I would like to point out to the authors that a correction based on the difference between observed and reference values for a single standard might not be enough, as recently highlighted out in Weng et al. 2020. Under the perspective of the influence of both humidity and isotopic composition, the correction function should represent a plane and not a line (as one could expect from Figure 4). Moreover, from Figure 4 it is not clear if the scatterplots reports the difference between a single standard (obs-ref) or for different standards (with different isotopic composition) because the plural "standards" word was used. Maybe the authors could highlight that under low humidity and low isotopic

variability, the humidity response of the instrument can be determined by analysing one (or two) standard(s).

Indeed, we paid attention to this calibration issue and ran several standards. For standards with low $\delta^{18}O$ values (-30 to -55 ‰) used for calibrating our instrument in Antarctica, the relationship between $\delta^{18}O$ ($\delta D$) and humidity is the same which makes the calibration of the laser analyzer easier in this region. On the contrary, when we use a less depleted standard ($\delta^{18}O$ =-9 ‰, $\delta D$ = -43.6 ‰), the relationship between $\delta D$ and humidity is not the same as displayed on Figure R2.

[Figure]

**Figure R2**: Relationship between humidity and $\delta^{18}O$ or $\delta D$ (when corrected from the reference value, i.e. the value of the standard measured at 2000 ppmv). VSMOW-VSLAP calibrated values of the standard used for this plot: EPB ($\delta^{18}O$ = -6.24 ‰; $\delta D$ = -43.6 ‰), NEEM ($\delta^{18}O$ = -33.50 ‰; $\delta D$ = -257.2 ‰), FP5 ($\delta^{18}O$ = -48.33 ‰; $\delta D$ = -383.5 ‰).

**Reviewer 2:**

**Remarks**
2.1 Evaporation Model
The theoretical foundation of the vapor generator presented here is described in section
2.1 of the manuscript. It is based on the assumption that a constant flow of water
ending on a syringe needle tip combined with a finely controlled flux of dry air will produce
an isotopically stable stream of vapor via a zero fractionation process. A prerequisite
for this, is that the size of the droplet remains stable throughout the experiment. It is
still a question to me how a droplet that sustains its size by constantly loosing mass on
its surface (regained by the incoming liquid water flow) via evaporation yields a vapor
stream that has the same isotopic composition as the liquid. This is a typical Rayleigh
evaporation likely with a strong kinetic component due to the very low humidity of the
carrier gas and the quick -but incomplete- removal of water molecules from the droplet.
Even though the treatment of the model has been presented in previous works, this is
so central to this system that at least an appendix with more information is needed.

We thank the reviewer for raising the question as it is indeed at first sight not immediately obvious
that the we can be certain that no fractionation factor is involved in the determination of the isotopic
composition of the output vapor. We have therefore decided to submit a companion paper that
describes and validates a model of the dynamic behavior of the humidity generator (Kerstel, AMTD,
2020). This manuscript is still in the reviewing process. In short, the mass balance model convincingly
demonstrates that the isotopic composition of the generated vapor is identical to that of the incoming
liquid, in steady-state. It also shows that in steady-state the droplet volume is constant and that the
enrichment occurring at the surface of the droplet due to evaporation of its surface water is
compensated by an inward diffusion of heavy isotopologues. This diffusion process, however, is frozen
in place by the continued renewal of water arriving in the droplet through the syringe needle. This is
important, as otherwise the bulk syringe water would become enriched. Measurements performed on
the syringe water before and after several hours of operation show that this is indeed not the case, as
predicted by our model.

We will add a brief discussion to this issue in the text and refer to the companion paper for more
details and insight. This also addresses the reviewer's request for an appendix, which we have thus
decided to replace by the companion paper.

2.1.1 Some points to consider
• How is the size of the droplet controlled when there is no camera or some other
monitoring tool inside the evaporation chamber?

The humidity level generated by the device is controlled in feed forward by the ratio of water to air
flow. The droplet size simply follows. There is thus no need to actually measure the droplet size. This
said, an early prototype contained a camera that allowed us to produce the photograph in Figure 1 in
the initial manuscript.

• The manuscript mentions in Line147 that the fractionation factor by Cappa et al
(2003) are used. But why if there is no fractionation?

We thank the reviewer for pointing out that the mention of Cappa, or other fractionation factors, is
superfluous here.

We will eliminate the mention of Cappa et al. in the revised version of the manuscript, and instead
refer to the companion paper for a discussion of the dynamic behavior of the device.

• The manuscript claims that under steady-state the isotopic composition of the generated vapour does not depend on the injected flux of water nor the specific humidity. Since this is a technical paper I would appreciate a simple experiment where the flux of the carrier gas is ramped up and down allowed to reach steady state. I have a very hard time seeing how this experiment will produce a vapor stream of the same isotopic composition equal to the $_{18}O$ of the liquid water. Similarly if the specific humidity of the carrier gas is changed one ends up with a simple mixing experiment (Mook , 2000) where the resulting vapour isotopic composition naturally will be different.

See previous answers for this point. We also followed the suggestion of the reviewer and run experiment with different dry air flows. Because the air is dry, it is not expected to modify the isotopic composition of the produced water vapor if the humidity is the same. This is exactly what is observed in Table R1.

• Throughout the whole manuscript, I have not seen a note on the temperature of the evaporation. This is a critical parameter affecting the efficiency of the evaporation, the saturation vapour pressure of the mixture and the fractionation factors in the (Cappa et al , 2003) parameterization. Since there is no active control and regulation of the temperature, all these parameters will vary.

We thank the reviewer for bringing up this issue, as the temperature is indeed a factor to consider for several reasons. Temperature variations affect the stability of the device by their effect on the stability of the flow regulator, which in turn affects the stability of the humidity level. Our flow regulator having a temperature sensitivity of 0.05 %/°C, this factor is deemed not critical. Temperature also intervenes through its effect on fractionation and the evaporation rate, as mentioned by the reviewer. Therefore, it can force a departure from steady-state operation, which in turn would drive isotope variations. For these reasons, the device is carefully insulated, leading to a temperature of the evaporation chamber of about 20°C varying within 1°C (over 24h).

2.2 Presentation–Standard nomenclature
Any physicist or engineer with moderate training in laboratory instrumentation should be able to look into the flow diagram of a manuscript in AMT or any other equivalent journal and get a basic idea of the method described in it. It is a very important element of a publication of this type, therefore it is my view that the authors should use standard P&ID nomenclature. The current flow diagram is a collection of coloured boxes from which little can be understood. Part of the text uses the photograph supplied (Figure 3) to explain the system something that confuses the reader even more.

2.2.1 Points to consider

• There are standard P&I symbols for valves, pumps, gauges etc that should be used, while colour should be added only if it aids in better explaining the system and not cosmetically. Standard names for the components should be used. For example three pumps with three different descriptions (pressure pump, pump and picarro pump) are found in the block diagram when they (most likely) are of the very same type ie vaccum pump.

The scheme initially presented on Figure 2 has been drawn again as presented in Figure R3 below.

[Figure]

**Figure R3:** Scheme of the humidity generator to replace figure 2 of the initial manuscript.

• One of the main elements of the paper according to the authors is the "double 3-way valve" added to the system. The valve's type is a 6-port switching valve and a standard symbol for it exists that should be used in this manuscript. It is central to describing the cycles of the system. Feel free to use the supplied (arguably not perfect) Fig. 1 that I am including and describes the valve operation for the syringe filling position.

[Figure]

**Figure 1**

As suggested by the reviewer, we now use the 6-port valve symbol in figure R3.

• Flows and pressure set points for the regulators also belong to the P&I diagram and since the authors claim that they are important to the operation of the system should be given. A table with the ID numbers of the control units and their set points would be very much appreciated. Currently Table 1 blends names of components from Figure 2 and Figure 3 so the reader has to guess. This is not informative.

In order to better document the system, we propose to add the following tables R2 and R3.

| Instruments | Notation on Figure R3 | Set points | Accuracy |
|---|---|---|---|
| Vögtlin GSC-A9TS-DD22 | FC | 300 and 150 sccm | 3.3 sccm |
| Harvard Apparatus Pump 11 Pico Plus Elite Dual | Syringe pump | 0.01 to 0.3 µL/min using 100 µL syringes | 0.35% of the set speed |
| Hamilton syringes | A and B | 100 µL | |
| Swagelok Ultra-Torr SS-4CD-TW-25 | Evaporation chamber A and B | Internal volume of 25 cm$^3$ | |
| Bronkhorst P-702CV-1K1A-AAD-22-V | P | 650 to 950 mbar | 3 mbar |
| KNF N86KNDC 24V | Pressure pump | | |

**Table R2**: Description and setting points of the instruments

| Humidity (ppmv) | Infusion rate (µL/min) | Dry Air flow (sccm) |
|---|---|---|
| 80 | 0.01 | 300 |
| 160 | 0.02 | 300 |
| 320 | 0.04 | 300 |
| 800 | 0.1 | 300 |
| 1200 | 0.15 | 300 |
| 1600 | 0.2 | 300 |

**Table R3:** Set points for water infusion rate and dry air flow at a temperature of 20°C

2.3 Experimental System–Explanation of operation
Section 2.2 of the manuscript contains information on the principle of operation. One of the puzzling pieces of information in this section is the carrier gas flow. Following the block diagram and the information in Table 1, we see that for the "Drying Mode", the Flow Controllers A and B sustain each a 400 sccm⁻¹ flow through the system. The block diagram does not indicate any open splits and for this particular mode the inlet valve is closed and the exhaust vacuum pump is disabled. This is an immense flow for the standard of a picarro spectrometer that normally can sustain its cavity pressure with sample flows in the order of 30-100 sccm⁻¹.

Similarly, during the injection of Standard A or Standard B, the flow is in the order of 450 sccm⁻¹, a very high flow level. In this mode though the exhaust vacuum pump is enabled, however we see that the lines of Standard A and Standard B are connected. Thus in Standard A mode, valve B is closed and the generated vapour from standard B, hopefully is evacuated via the exhaust vacuum pump. Is the pressure of 905 mbar enough to make sure that all the 150 sccm⁻¹ of standard B are off the system and not mixed with the vapour from Standard A? Why was this value chosen and what kind of tests assure that the two lines are never mixed even though physically connected. The safe way to proceed here would be to simply isolate the two channels, remove the pressure regulator and the vacuum pump and simply install two 3-way valves in the position of valves A and B (which I assume are On/Off???) on which one port goes to the instrument inlet and the other is simply open to the atmosphere in an open split configuration.

2.3.1 Points to consider

• A more thorough description of the flow path is needed and the issue of possible
cross contmination between Standard A and B should be addressed.

The instrument has been designed with tubing long enough to prevent any mixing and the issue of
mixing has been extensively tested during conception by comparing the isotopic composition of the
produced water vapor when only one way was in operation (bottle A filled with standard and bottle
B empty) and when two ways were in operations (bottles A and B filled with standards of different
isotopic composition). the isotopic composition produced on way A was always the same.
To double check this effect when receiving this review, we performed an additional comparison:

- Case A: bottle B was filled with FP5 standard ($\delta^{18}O$ = -48.33 ‰) and bottle A filled with EPB
  standard ($\delta^{18}O$ = -6.24 ‰). Measured d18O on way B was -48.96±1‰ (humidity 768±5
  ppmv)
- Case B: bottle A and B were filled with the FP5 standard. Measured $\delta^{18}O$ on way B was -
  49.05±1‰ (humidity 765±4 ppmv).

There is thus no difference between the two cases both for $\delta^{18}O$ and $\delta D$ which confirms that there is
no mixing in the system. We will explain these tests and the absence of mixing effect in the revised
version of the manuscript.

• The flows are very high - How does the picarro cope with this condition?

There is a waste line on this instrument (corresponding to the outlet of the pressure pump) so that the
Picarro laser analyzer is only pumping what it needs through valves A and B as it does during the
analysis of atmospheric air through inlet valve. As a consequence, there is no problem for the system.
This will be mentioned in the revised version of the manuscript.

• There is no information on the volume of the evaporation chambers and no mention
regarding the temperature of the system.

Volume of the evaporation chamber is 25 cm$^3$ (see reference of the chamber on Table R2).
Temperature is around 20°C.

• There are references to a two standard calibration protocol in this section. It is
not exactly clear if this concerns some water concentration correction calibration
or a linear slope SMOW-SLAP calibration–see specific comment on the notable
absence of any reference to SMOW-SLAP below.

All the lab-standards used in this study are indeed calibrated versus SMOW using a 3-point
calibration which will be mentioned in the revised version of the manuscript. Then, we used two
ways for our generator in order to perform two-point calibrations of the laser analyzer when it is
used to measure water vapor isotopic composition at low humidity. We hope that the answers
provided for the other comments below now explain better this aspect and it will also be
implemented in the revised manuscript. This is indeed very important to mention.

• I assume that the system is also used in order to perform a SMOW-SLAP calibration.
How are evaporation/fractionation effects in the standards' vials handled?
Based on Figure 3 we see no precautions concerning this.

Thank you for this comment. Indeed, this was not mentioned in the manuscript. This has been tested by sampling the water from the flasks every 2 weeks. We observed a maximum evolution of the isotopic composition of the standard in the bottle by 0.05 ‰ for $\delta^{18}O$ and 0.5 ‰ in $\delta D$ within 2 months in case of intense use of the instrument. This will be mentioned in the new version of the manuscript.

2.4 Evaluation–Measurement Stability and precision
The manuscript lacks important information on the evaluation of the performance of
the system. Section 3.1 assures the reader that no fractionation is observed during the
generation of the vapour and its transfer in the flow lines while the agreement with the
commercial standard delivery module is within 0.2 and 1‰ for $_{18}O$ and $_D$ respectively.
It is almost impossible to judge these discrepancies based on figure 5 and in fact
when looking closely it is rather difficult to see how the differences between the two
systems are of this order.

With respect to figure 4 and the evaluation of the precision for [H2O] and $_{18}O/_D$ I
strongly believe that the manuscript should include a proper Allan variance test (Werle
et al , 2011; Steig et al , 2014). It is obvious that the system experiences drifts, whose
origing is the vapour generator. Calculating the standard deviation on a 10 min window
of a time series that obviously has a changing mean value looks and feels like possible
cherry-picking.

2.4.1 Points to consider–Suggestions
• Show better evidence of zero fractionation. If you choose the SDM as a measure
of comparison show a detailed comparison and plot the results clearly. The axes
of fig 5 span 300 ‰ for $_D$. It is impossible to say anything.

In addition to the current text which already mention the detailed comparison on l. 250 to 253, a focus on the comparison between the SDM and the low-humidity level generator is displayed on Figure R4 below. This figure will be implemented in the revised manuscript.

[Figure]

**Figure R4**: Comparison of the difference in isotopic composition ($\delta D$ top, $\delta^{18}O$ bottom) between the measured and the reference (V-SMOW calibrated) values as obtained by the SDM (red) and the low-humidity level generator (black) coupled to the same Picarro L 2130-i with the same lab-standard (FP5).

• Calculate the Allan variance of the system for 4-5 [H2O] levels. Let one water run

for several hours and so you get a more honest idea about precision and drifts for all three parameters under consideration. It is a very simple experiment that can provide a valuable insight and by looking in the bibliography it is a rather standard evalluation tool for laser spectroscopy based measurements.

Thank you for the suggestion. Indeed, this is a good idea and we propose to include the figure R1 (see answer to reviewer 1) in the revised manuscript

2.5 Water concentration correction

The water concentration correction is described in section 3.3 with Figure 5 being the main source of information for this part of the analysis. I would consider this one of the most important sections in the paper and unfortunately it spans only one paragraph. It is repeated in the manuscript that the system presented here is superior to the commercial standard delivery module below the level of 500 ppm. How can we see this when the SDM measurements in this are of [$H_2O$] are not presented (measurements stop at [$H_2O$]_2000 ppm)?

It is very difficult to get stable humidity plateau with a SDM below 500 ppmv. This is the reason why we could not display them. This will be mentioned in the new manuscript.

The dataset in Figure 5 presents the difference of the raw isotopic value and a value that the authors call "reference" and "real". These words commonly refer to very specific things in isotope geochemistry and my guess is that the reference value is a SMOW-SLAP calibrated value (?). The term SMOW-SLAP calibration is not to be found at all in the manuscript. It is to some extent problematic that a water concentration correction measurement compares a raw value at a number of [$H_2O$] levels with a value post SMOW-SLAP calibration. Please see my comments on SMOW-SLAP in the next section.

As mentioned above, the SMOW-SLAP calibration will be addressed in the revised manuscript and all our lab-standards are calibrated vs V-SMOW (using also V-SLAP as a reference).

The type of correction is not described in the manuscript. This is what this work is all about. There can be various approaches on how it can be done (see for example (Schmidt et al , 2010; Gkinis et al , 2010; Aemisegger et al , 2012)) but the authors need to be open and specific about what they did exactly. Moreover (Schmidt et al , 2010) suggest that the response to changing [$H_2O$] levels varies with the isotopic composition. I do not know how important this effect is for this present work, but I believe that the authors should perform two separate experiments with two different waters (technically the system offers this possibility). It would also strengthen their claims for a vary stable system if they show the full experiment with its raw data as a function of time and not only the averages.

There is also a claim in this section that the stability of the system allows via the daily measurement of standards waters to quantify instrumental drifts. There is no strict evaluation of the stability of the system so far and I think that a proper Allan variance test at various levels of [$H_2O$] is necessary. But even in the absence of an Allan variance test the data given in Table 2 do not necessarily indicate instrumental drifts. All the isotope data in the table are within 1 ⊡ _ of the noise level for a 10-min period mentioned in the caption and there does not seem to be a one way trend. So how is it possible to say anything about instrumental drifts? Also the noise levels

mentioned in the caption of Table 2 are very different for roughly the same [H2O] when
one looks at the data of Figure 4 (4.5 versus 14).

This is a good remark indeed and we are sorry for the confusion. Actually, while the noise on the humidity level is mainly due to the low-humidity level generator, the noise on the isotopic ratios depends on the laser analyzer which explains some inconsistency in the results which were displayed (we presented results obtained with different instruments). The inconsistencies will be fixed in the revised version. Indeed, Table 2 shows that there is no measurable drift and this will be explained more clearly in the revised version.

2.5.1 Points to consider-Suggestions
• A clear description of the correction is missing. Some math is needed here.

We propose to provide the following explanation in the revised manuscript.

In order to correct the dependence of the isotopic composition ($\delta^{18}$O, $\delta$D) to humidity, we use the LHLG with two lab-standards with isotopic composition ideally bracketing the isotopic composition of the water vapor measured by the analyzer. In our case, we used our two lowest lab-standards and obtained for the two standards the same dependency of isotopic composition vs humidity (in the case of Antarctica, we use the NEEM and FP5 standards, Figure R2). We express this dependency as the relationship between the difference in $\delta$D or $\delta^{18}$O between the measured value at the given humidity and the value of the same standard measured at a humidity of 2000 ppmv. The experimental data from figures 5 andR2 are fitted through a polynomial function with respect to humidity h (in ppmv):

$\delta^{18}$O $- \delta^{18}$O$_{ref}$ = 0.0000000000000000397×h$^6$ - 0.00000000000003586315×h$^5$ + 0.0000000012843645994×h$^4$ - 0.0000023087753445094×h$^3$ + 0.00021857285350473100×h$^2$ - 0.10603325432255400000×h + 23.7  (eq. 1)

$\delta$D $- \delta$D$_{ref}$ = 0.00000000000000006859×h$^6$ - 0.00000000000060047709×h$^5$ + 0.0000000207903313490×h$^4$ - 0.0000361319302207374×h$^3$ + 0.00330716141498371000×h$^2$ - 1.53651645114701000000×h + 313 (eq. 2)

A second important correction step is to take into account the slope of the VSMOW calibrated $\delta$D and $\delta^{18}$O on the Picarro laser analyzer vs humidity-corrected measured $\delta^{18}$O and $\delta$D. Depending on the instrument used, the slope lies between 0.96 and 1, cf Table R4 below for an example.

• An experiment at two different isotopic levels showing the raw data versus time
will show if there is an isotope effect in the water concentration dependence.

Thanks for this suggestion. We will display the tests performed in figure R2 (see also corresponding answer to reviewer 1).

• A proper quantification of instrumental drifts (and this concerns the system as a
whole and not only the spectrometer) can be done with a proper Allan variance
test.

This is the aim of figure R1 which will be implemented in the revised version of the manuscript.

2.6 Lack of reference to SMOW-SLAP calibration
The main goal of building a water vapour generator as a peripheral for isotope measurements
of water vapour is to be able to calibrate the dataset on the SMOW-SLAP

scale. This is the only way to communicate and compare the measurements with other existing data sets and produce some science out of them. It is also even more important if the deuterium excess parameter will be studied as it is very sensitive to this calibration procedure. Therefore it appears very awkward that a manuscript dealing with this topic does not include a single note, comment or reference to this very important step Of the measurement process. The dataset presented later in section 4 of the manuscript are impossible to evaluate if they are not calibrated in the SMOW-SLAP scale.

One more purpose of performing such calibrations, is that they can reveal possible accuracy issues in the instrumentation system. Given two standard waters one should be able to produce a calibration line and thereafter measure a third water of known isotopic composition treating it as an unknown. If the resulting value lies beyond the 3 ▢ _ range then there is likely something wrong with the system. That could be any part from the water standard storage to the water vapour generation system or the spectrometer itself. Currently there is no way to say anything about the accuracy of the system. With this in mind, section 4 of the manuscript is of very little use as the dataset is reported on some local instrument scale.

A SMOW-SLAP calibration experiment at various [H2O] levels using the SDM and the current system would provide a proper comparison between the two systems and therefore it would be a very important addition to the manuscript.

2.6.1 Points to consider
• A proper treatment of the SMOW-SLAP calibration step is notably missing.

We will explain in the revised manuscript and in the caption of figure 6 hat all our lab-standards are regularly calibrated vs SMOW using IAEA VSMOW and VSLAP standards (every 3 years in minimum). In order to calibrate our lab-standards, we use determination from Picarro L 2130-i for $\delta D$ and for $\delta^{18}O$. In addition, we use a $\delta^{18}O$ calibration using an IRMS through water – $CO_2$ equilibration.

• Performing 2-standard calibrations and measuring a third water standard treated as an unknown will be a valuable -almost essential- addition to the manuscript, offering important information on the accuracy of the system.

This is an excellent suggestion and this test has been performed as follows:

| Standard | Calibrated value (VSMOW – VSLAP) | Measured value at 800 ppmv | Measured value corrected from humidity dependence (Equation 1) |
|---|---|---|---|
| EPB | -6.24 ‰ | -8.27 ‰ | -10.78 ‰ |
| NEEM | -33.5 ‰ | -34.48 ‰ | -36.99 ‰ |
| FP5 | -49.11 ‰ | -49.02 ‰ | -51.53 ‰ |

**Table R4**: Comparison of measured vs VSMOW-VSLAP calibrated $\delta^{18}O$ values for 3 standards measured with a Picarro analyzer after generation of water vapor using the low-level humidity generator.

We used the measured and true values of EPB and FP5 to estimate the $\delta^{18}O$ value of the NEEM standard from its measured value. Using the linear relationship obtained from VSMOW-VSLAP calibrated EPB and FP5 $\delta^{18}O$ vs measured EPB and FP5 $\delta^{18}O$ values ($\delta^{18}O_{cal} = 1.0329*\delta^{18}O_{meas} + 4.8965$) leads to an estimated NEEM $\delta^{18}O$ of -33.31 ‰.

Another approach because EPB and FP5 $\delta^{18}O$ are very different, is to scale the isotopic composition using the isotopic ratios (1+$\delta$) so that:

$(1+\delta^{18}O_{cal, NEEM}) = (1+ \delta^{18}O_{meas,NEEM})/(1+\delta^{18}O_{meas,EPB})*$
$[(1+\delta^{18}O_{cal,FP5})/(1+\delta^{18}O_{cal,EPB})]/[(1+\delta^{18}O_{meas,FP5})/(1+\delta^{18}O_{meas,EPB})]*(1+\delta^{18}O_{cal,EPB})$

With this approach, the estimated NEEM $\delta^{18}O$ value is -33.74 ‰.

Given the uncertainty of about 0.8-1 ‰ when measuring $\delta^{18}O$ around 800 ppmv, we can conclude that the system is accurate.

• Since a lot has been written about the performance of the commercial SDM it will
be proper to perform 2-standard calibrations for various [H2O] levels and compare
the results.

See answer above as well as the Figure R4.

We also propose to display the comparison between SDM and the low-humidity level generator (Figure R5) which shows the high stability of humidity with the new instrument compared to the SDM.

[Figure]

[Figure]

**Figure R5**: Comparison of humidity plateaus (800 ppmv) generated with the low-level humidity generator (top) and with the SDM (bottom).

**References:**

Kerstel, E.: Modeling the dynamic behavior of a droplet evaporation device for the delivery of isotopically calibrated low-humidity water vapor, Atmos. Meas. Tech. Discuss. [preprint], https://doi.org/10.5194/amt-2020-428, in review, 2020.

Landsberg, J.: Développement d'un spectromètre laser OF-CEAS pour les mesures des isotopes de la vapeur d'eau aux concentrations de l'eau basses. [online] Available from: http://www.theses.fr/2014GRENY052/document, 2014.

Merlivat, L., Molecular diffusivities of H216O, HD16O and H218O in gases, J. Phys. Chem., 69(6), 2864–2871, 1978.

---

## Author Response (AR1)

We thank the reviewers for very useful comments. We provide below the answers to the different comments and revised the manuscript accordingly. Note that part of the answers as well as the need for a more detailed explanation of the theory associated with the vaporization of the drop is provided in a companion paper which is now available online (Kerstel, AMTD, 2020). We also added one co-author, Morgane Farradèche, who realized numerous of the additional tests requested by the reviewers.

**Reviewer 1**

Technical comments: 1- This study represents an extension of the work shown in Landsberg (2014). However, in his study a strong influence of the lead screw rotation was identified as one of the main source of noise in the water vapor signal at low humidity. Is the choice of the pump a critical point for the development of the instrument? There are other critical points for designing/replicating this device that must be taken into account (e.g. how important is the choice mass flow and pressure controller? The authors should discuss this point e.g. in section 2.2.

Indeed the choice of pump is important. First of all because it needs to provide an exceedingly small water flux, and secondly because it needs to do so with high stability. It is therefore noted that the oscillations that are in-phase with the lead-screw rotation were seen by Landsberg (2014) in an early prototype and were eliminated by better engineering, and in particular by the careful elimination of air-bubbles in the liquid water supply lines and syringe. This aspect is also addressed in our companion paper (Kerstel, 2020).

The precision of the pressure controller is not critical. Its purpose is to provide a steady inlet pressure to the optical spectrometer, also during syringe switching. Ideally the spectrometer is not sensitive to the inlet pressure.

The precision of the flow controller on the other hand does directly influence the precision of the volume mixing ratio (humidity level) produced by the instrument. Its precision ranges from about 0.5% at the highest flow setting, to almost 50% at the lowest air flow setting. In practice the device is used with an intermediate air flow at which the precision of the controller is specified to be 1%. We observe, however, that in practice its short-term precision is much better. In any event, a precision of the order of 1% is normally comparable to the precision of the measurement of the humidity level with the optical spectrometer.

The text has been modified accordingly to provide quantitative information on these issues and cite the companion paper.

Section 2.2

"The spectrometer is not sensitive to the inlet pressure, the precision of the pressure controller is not an essential aspect. On the contrary, the precision of the flow controller is key for the precision of the humidity level produced by the instrument: it is of 1% for the air flow which is comparable to the precision of the measurement of the humidity level with the optical spectrometer."

2- Since large stability of water vapor flux is expected by such device, one would expect an analysis of the stability of water vapor signal, at least in terms of the mixing ratio. An analysis of stability could be the identification/absence of trend during different humidity steps or the analysis of mixing ratio standard deviation as a function of different instrumental configuration (e.g. dry air flux, syringe speed etc). A stability analysis would provide additional evidence of the robustness/reliability of the instrument.

This issue was addressed through two different tests. First, we now present Allan variance obtained over 4 hours at different humidity levels by varying the syringe speed (Figure 3 of the new manuscript)

[Figure]

(a)                                    (b)                                    (c)

**Figure 3**: Allan variance over 4 hours for different humidity levels (black 1080 ppmv; blue 770 ppmv; green 400 ppmv; yellow 320 ppmv) for humidity (a), $\delta^{18}O$ (b) and $\delta D$ (c).

Second, we also tested the stability of the system by varying the flow of dry air from 200 to 400 sccm (Table S2 of the manuscript). The reported standard deviation is over a period of 10 minutes (last 10 minutes of the plateau) but the humidity and $\delta^{18}O$ values were stable over one hour (the experiment was stopped after one hour of stable humidity). We notice that the generated $\delta^{18}O$ values does not depend on the dry air flow nor on the infusion rate.

| Air flow (sccm) | Infusion rate (µL/min) | Humidity (ppmv) | stdev humidity over 10 minutes (ppmv) | $\delta^{18}O$ (‰) | stdev $\delta^{18}O$ over 10 minutes (‰) |
|---|---|---|---|---|---|
| 200 | 0.07 | 808 | 1 | -7.88 | 0.89 |
| 300 | 0.11 | 851 | 2 | -7.73 | 0.85 |
| 400 | 0.14 | 818 | 2 | -7.95 | 0.90 |
| 200 | 0.03 | 374 | 1 | -8.45 | 1.92 |
| 300 | 0.05 | 411 | 2 | -9.16 | 1.64 |
| 400 | 0.07 | 415 | 3 | -9.05 | 1.59 |

**Table S2**: Evolution and stability of humidity and $\delta^{18}O$ (same water used for the different tests) for different syringe infusion rates and dry air flows.

Other minor comments

L136 Period.

Done

L147-148 What is the reason for using fractionation factors of Cappa et al. (2003)?

In this paper we are interested in the steady-state behavior of the device. In this case, the isotopic composition of the incoming liquid water is by mass balance considerations equal to the isotopic composition of the evaporated water flux. The fractionation factor between the liquid and vapor phase is thus of no importance for the work presented here. The previously mentioned companion paper,

however, deals with the dynamic behavior of the humidity generator and discusses the fractionation factors in detail. There we use estimations of the fractionation factors that are based on the studies by Cappa et al. (2003) and Luz et al. (2009). The work concludes that the effective fractionation factors are largely determined by their equilibrium counter parts, already determined with high precision by Horita and Wesolowski (1994), combined with the ratio of diffusivities and a parameter that describes the flow regime (from laminar to slightly turbulent). The diffusivities were determined by Merlivat (1978), Cappa et al. (2003) and more recently by Luz et al. (2009). We used the values by Cappa et al. (2003), but the difference is minimal for our purpose. Cappa et al. (2003) predicts only slightly lower values of the effective fractionation factor in the laminar limit compared to other estimates.

As the fractionation factors are not necessary for this steady-state study, we will remove the reference to Cappa in this study. The dynamical aspects are dealt in the companion paper.

Figure 2. Check part labels (A/B) because different names are used in text, in table 1 (F1, G1 etc..) and in Figure 3.

The scheme has been fully changed, see answer to reviewer 2 and new Figure 2 and Table S1.

L245 In relationship with my technical comment #2: what stable means? No trend observed in mixing ratio? Low RSD?

Indeed, stable means both no trend in the mixing ratio (< 20 ppmv/hour) and low RSD (< 10 ppmv over 10 minutes). This is explained in the revised manuscript.

"The LHLG is able to generate stable levels of humidity (drift lower than 20 ppmv over one hour and $1\sigma$ below 10 ppmv over 10 minutes)"

Figure 4. It is not clear how long it takes the signal to stabilize and whether this stabilization period is related to the humidity level. From Fig.4 it seems so, because injection steps are characterised by different lengths.

Actually, it can be variable between a few minutes only to 30 minutes in worst cases. In best case, we have stabilization in a few minutes but it may take more time for undetermined reason. This is the reason why, in routine mode (i.e. when the user is not checking visually the stability of the plateau), we run 50-minute long plateau. This is explained in the revised text.

"In the routine mode (Figure 4), we perform plateaus of 30 to 50 minutes (50 minutes when the instrument is unattended since the time to reach the plateau varies between a few minutes to 30 minutes)."

Section 3.2 Maybe, two words here (or a simple scatter plot, boxplot) about a possible relationship between precision and humidity level could help the reader understand the stability of the system.

There is actually no relationship between precision of humidity measurement and humidity level as seen in figure 3a (new in this manuscript).

Section 3.3, Figure 5 and Section 4: humidity correction is important. I would like to point out to the authors that a correction based on the difference between observed and reference values for a single standard might not be enough, as recently highlighted out in Weng et al. 2020. Under the perspective of the influence of both humidity and isotopic composition, the correction function should represent a plane and not a line

(as one could expect from Figure 4). Moreover, from Figure 4 it is not clear if the scatterplots reports the difference between a single standard (obs-ref) or for different standards (with different isotopic composition) because the plural "standards" word was used. Maybe the authors could highlight that under low humidity and low isotopic variability, the humidity response of the instrument can be determined by analysing one (or two) standard(s).

Indeed, we paid attention to this calibration issue and ran several standards. For standards with low $\delta^{18}O$ values (-30 to -55 ‰) used for calibrating our instrument in Antarctica, the relationship between $\delta^{18}O$ ($\delta D$) and humidity is the same which makes the calibration of the laser analyzer easier in this region. On the contrary, when we use a less depleted standard ($\delta^{18}O$ =-9 ‰, $\delta D$ = -43.6 ‰), the relationship between $\delta D$ and humidity is not the same as displayed on the new figure 5.

[Figure]

*__Figure 5__: Influence of humidity on the isotopic composition ($\delta^{18}O$ and $\delta D$) of the vapor obtained with the LHLG with 3 water lab-standards. The error bars are calculated as the standard deviation ($1\sigma$) over the generated values by the L2130-i instrument during 10 minutes at 1 second resolution (i.e. without any pre-averaging of the raw dataseries). The $\delta^{18}O_{ref}$ and $\delta D_{ref}$ are the values of the injected water standards at 2,000 ppmv.*

**Reviewer 2:**

**Remarks**

2.1 Evaporation Model

The theoretical foundation of the vapor generator presented here is described in section 2.1 of the manuscript. It is based on the assumption that a constant flow of water ending on a syringe needle tip combined with a finely controlled flux of dry air will produce an isotopically stable stream of vapor via a zero fractionation process. A prerequisite for this, is that the size of the droplet remains stable throughout the experiment. It is still a question to me how a droplet that sustains its size by constantly loosing mass on its surface (regained by the incoming liquid water flow) via evaporation yields a vapor stream that has the same isotopic composition as the liquid. This is a typical Rayleigh evaporation likely with a strong kinetic component due to the very low humidity of the carrier gas and the quick -but incomplete- removal of water molecules from the droplet. Even though the treatment of the model has been presented in previous works, this is so central to this system that at least an appendix with more information is needed.

We thank the reviewer for raising the question as it is indeed at first sight not immediately obvious that the we can be certain that no fractionation factor is involved in the determination of the isotopic composition of the output vapor. We have therefore decided to submit a companion paper that describes and validates a model of the dynamic behavior of the humidity generator (Kerstel, AMTD, 2020). This manuscript is still in the reviewing process. In short, the mass balance model convincingly demonstrates that the isotopic composition of the generated vapor is identical to that of the incoming liquid, in steady-state. It also shows that in steady-state the droplet volume is constant and that the enrichment occurring at the surface of the droplet due to evaporation of its surface water is compensated by an inward diffusion of heavy isotopologues. This diffusion process, however, is frozen in place by the continued renewal of water arriving in the droplet through the syringe needle. This is important, as otherwise the bulk syringe water would become enriched. Measurements performed on the syringe water before and after several hours of operation show that this is indeed not the case, as predicted by our model.

We will add a brief note to this issue in the text and refer to the companion paper for more details and insight. This also addresses the reviewer's request for an appendix, which we have thus decided to replace by the companion paper.

"By solving numerically the differential equation (2), it is possible to faithfully simulate the behavior of the device under changing conditions (Kerstel, 2020). This numerical approach validates the theoretical explanation of the undersaturated evaporation of the droplet. Importantly, it is noted that in steady-state, the isotopic composition of the generated humid air is identical to that of the injected water stream, and therefore does not depend on the infusion rate, nor on the specific humidity."

2.1.1 Some points to consider
• How is the size of the droplet controlled when there is no camera or some other monitoring tool inside the evaporation chamber?

The humidity level generated by the device is controlled in feed forward by the ratio of water to air flow. The droplet size simply follows. There is thus no need to actually measure the droplet size. This said, an early prototype contained a camera that allowed us to produce the photograph in Figure 1.

• The manuscript mentions in Line147 that the fractionation factor by Cappa et al (2003) are used. But why if there is no fractionation?

We thank the reviewer for pointing out that the mention of Cappa, or other fractionation factors, is superfluous here.

We eliminated the mention of Cappa et al. in the revised version of the manuscript, and instead refer to the companion paper for a discussion of the dynamic behavior of the device.

• The manuscript claims that under steady-state the isotopic composition of the generated vapour does not depend on the injected flux of water nor the specific humidity. Since this is a technical paper I would appreciate a simple experiment where the flux of the carrier gas is ramped up and down allowed to reach steady state. I have a very hard time seeing how this experiment will produce a vapor stream of the same isotopic composition equal to the $_{18}O$ of the liquid water. Similarly if the specific humidity of the carrier gas is changed one ends up with a simple mixing experiment (Mook , 2000) where the resulting vapour isotopic composition naturally will be different.

We followed the suggestion of the reviewer and run experiment with different dry air flows. Because the air is dry, it is not expected to modify the isotopic composition of the produced water vapor if the humidity is the same. The results are shown in the new manuscript on Table S2 displayed above as answer to reviewer 1.

• Throughout the whole manuscript, I have not seen a note on the temperature of the evaporation. This is a critical parameter affecting the efficiency of the evaporation, the saturation vapour pressure of the mixture and the fractionation factors in the (Cappa et al , 2003) parameterization. Since there is no active control and regulation of the temperature, all these parameters will vary.

We thank the reviewer for bringing up this issue, as the temperature is indeed a factor to consider for several reasons. Temperature variations affect the stability of the device by their effect on the stability of the flow regulator, which in turn affects the stability of the humidity level. Our flow regulator having a temperature sensitivity of 0.05 %/°C, this factor is deemed not critical. Temperature also intervenes through its effect on fractionation and the evaporation rate, as mentioned by the reviewer. Therefore, it can force a departure from steady-state operation, which in turn would drive isotope variations. For these reasons, the device is carefully insulated, leading to a temperature of the evaporation chamber of about 20°C varying within 1°C over 24h.

In the new text, we precise:
"Temperature intervenes through its effect on fractionation and the evaporation rate (apart from a negligible effect on the flow controller stability), which could lead to a departure from steady-state operation. For these reasons, the temperature of the evaporation chambers was maintained at 20 °C (within 1 °C over 24 hours). »

2.2 Presentation–Standard nomenclature
Any physicist or engineer with moderate training in laboratory instrumentation should be able to look into the flow diagram of a manuscript in AMT or any other equivalent journal and get a basic idea of the method described in it. It is a very important element of a publication of this type, therefore it is my view that the authors should use standard P&ID nomenclature. The current flow diagram is a collection of coloured boxes from which little can be understood. Part of the text uses the photograph supplied (Figure 3) to explain the system something that confuses the reader even more.

2.2.1 Points to consider

• There are standard P&I symbols for valves, pumps, gauges etc that should be used, while colour should be added only if it aids in better explaining the system and not cosmetically. Standard names for the components should be used. For example three pumps with three different descriptions (pressure pump, pump and picarro pump) are found in the block diagram when they (most likely) are of the very same type ie vaccum pump.

The scheme of Figure 2 has been drawn again as:

[Figure]

*Figure 2: Humidity generator schematic diagram (see supplementary Table S1 for details on the different elements)*

The same symbol is used for the different pumps but we still give a different name for each pump to help making a distinction between the different pumps in the text.

• One of the main elements of the paper according to the authors is the "double 3-way valve" added to the system. The valve's type is a 6-port switching valve and a standard symbol for it exists that should be used in this manuscript. It is central to describing the cycles of the system. Feel free to use the supplied (arguably not perfect) Fig. 1 that I am including and describes the valve operation for the syringe filling position.

[Figure]

Position 1
filling syringes

**Figure 1**

As suggested by the reviewer, we now use the 6-port valve symbol in the figure 2.

• Flows and pressure set points for the regulators also belong to the P&I diagram and since the authors claim that they are important to the operation of the system should be given. A table with the ID numbers of the control units and their set points would be very much appreciated. Currently Table 1 blends names of components from Figure 2 and Figure 3 so the reader has to guess. This is not informative.

In order to better document the system, we propose to add the following tables S1 and 2.

| Instruments | Notation on Figure 2 | Setting points | Accuracy |
|---|---|---|---|
| Vögtlin GSC-A9TS-DD22 | FC | 300 and 150 sccm | 3.3 sccm |
| Harvard Apparatus Pump 11 Pico Plus Elite Dual | Syringe pump | 0.01 to 0.3 µL/min using 100 µL syringes | 0.35% of the set speed |
| Hamilton syringes | A and B | 100 µL | |
| Swagelok Ultra-Torr SS-4CD-TW-25 | Evaporation chamber A and B | Internal volume of 25 cm$^3$ | |
| Bronkhorst P-702CV-1K1A-AAD-22-V | P | 650 to 950 mbar | 3 mbar |
| KNF N86KNDC 24V | Pressure pump | | |

**Table S1**: *Description and setting points of the elements composing the LHLG*

| Humidity (ppmv) | Infusion rate (µL/min) | Dry Air flow (sccm) |
|---|---|---|
| 80 | 0.01 | 300 |
| 160 | 0.02 | 300 |
| 320 | 0.04 | 300 |
| 800 | 0.1 | 300 |
| 1200 | 0.15 | 300 |
| 1600 | 0.2 | 300 |
| 2400 | 0.3 | 300 |

**Table 2:** *Setting points for water infusion rate and dry air flow at a temperature of 20°C.*

2.3 Experimental System–Explanation of operation
Section 2.2 of the manuscript contains information on the principle of operation. One of the puzzling pieces of information in this section is the carrier gas flow. Following the block diagram and the information in Table 1, we see that for the "Drying Mode", the Flow Controllers A and B sustain each a 400 sccm⁻1 flow through the system. The block diagram does not indicate any open splits and for this particular mode the inlet valve is closed and the exhaust vacuum pump is disabled. This is an immense flow for the standard of a picarro spectrometer that normally can sustain its cavity pressure with sample flows in the order of 30-100 sccm⁻1.

Similarly, during the injection of Standard A or Standard B, the flow is in the order of 450 sccm$^{-1}$, a very high flow level. In this mode though the exhaust vacuum pump is enabled, however we see that the lines of Standard A and Standard B are connected. Thus in Standard A mode, valve B is closed and the generated vapour from standard B, hopefully is evacuated via the exhaust vacuum pump. Is the pressure of 905 mbar enough to make sure that all the 150 sccm$^{-1}$ of standard B are off the system and not mixed with the vapour from Standard A? Why was this value chosen and what kind of tests assure that the two lines are never mixed even though physically connected. The safe way to proceed here would be to simply isolate the two channels, remove the pressure regulator and the vacuum pump and simply install two 3-way valves in the position of valves A and B (which I assume are On/Off???) on which one port goes to the instrument inlet and the other is simply open to the atmosphere in an open split configuration.

2.3.1 Points to consider

• A more thorough description of the flow path is needed and the issue of possible cross contmination between Standard A and B should be addressed.

The instrument has been designed with tubing long enough to prevent any mixing and the issue of mixing has been extensively tested during conception by comparing the isotopic composition of the produced water vapor when only one way was in operation (bottle A filled with standard and bottle B empty) and when two ways were in operations (bottles A and B filled with standards of different isotopic composition). The isotopic composition produced on way A was always the same. To double check this effect when receiving this review, we performed an additional comparison:

- Case A: bottle B was filled with FP5 standard ($\delta^{18}O = -48.33$ ‰) and bottle A filled with EPB standard ($\delta^{18}O = -6.24$ ‰). Measured $\delta^{18}O$ on way B was -48.96±1‰ (humidity 768±5 ppmv)
- Case B: bottle A and B were filled with the FP5 standard. Measured $\delta^{18}O$ on way B was -49.05±1‰ (humidity 765±4 ppmv).

There is thus no isotopic difference between the two cases both for $\delta^{18}O$ and $\delta D$ which confirms that there is no mixing in the system. These tests and the absence of mixing effect now constitute text S1 in the revised version of the manuscript:

**Text S1: No mixing between standards A and B during vaporization**

The instrument has been designed with tubing long enough to prevent any mixing. Still, the issue of mixing has been extensively tested during conception by comparing the isotopic composition of the produced water vapor when only one way was in operation (bottle A filled with standard and bottle B empty) and when two ways were in operations (bottles A and B filled with standards of different isotopic composition). The isotopic composition produced on way A was always the same. An additionnal comparison is displayed below.

- Case A: bottle B was filled with FP5 standard ($\delta^{18}O = -48.33$ ‰) and bottle A filled with EPB standard ($\delta^{18}O = -6.24$ ‰). Measured $\delta^{18}O$ on way B was -48.96±1‰ (humidity 768±5 ppmv)
- Case B: bottle A and B were filled with the FP5 standard. Measured $\delta^{18}O$ on way B was -49.05±1‰ (humidity 765±4 ppmv).

There is no difference between cases A and B which confirms that there is no mixing in the system.

• The flows are very high - How does the picarro cope with this condition?

There is a waste line on this instrument (corresponding to the outlet of the pressure pump) so that the Picarro laser analyzer is only pumping what it needs through valves A and B as it does during the analysis of atmospheric air through inlet valve. As a consequence, there is no problem for the system. This is now mentioned in the revised version of the manuscript.

"When the instrument is connected to the infrared spectrometer, the excess humid air flow is exhausted to the room through the pressure pump and the spectrometer only pumps what is required (Figure 2)."

• There is no information on the volume of the evaporation chambers and no mention regarding the temperature of the system.

Volume of the evaporation chamber is 25 cm$^3$ (see Table S1). Temperature is around 20°C (see above and in section 2.2).

• There are references to a two standard calibration protocol in this section. It is not exactly clear if this concerns some water concentration correction calibration or a linear slope SMOW-SLAP calibration–see specific comment on the notable absence of any reference to SMOW-SLAP below.

Indeed, this was missing in the previous manuscript. All the lab-standards used in this study are indeed calibrated versus SMOW using a 3-point calibration which will be mentioned in the revised version of the manuscript. Then, we used two ways for our generator (one way for each lab-standard) in order to perform two-point calibrations of the laser analyzer when it is used to measure water vapor isotopic composition at low humidity. We hope that the answers provided for the other comments below now explain better this aspect. The calibration protocol of our data is now explained with the addition of equations 1 and 2 as well as new section 3.5 given later in the answer to reviewer 2.

• I assume that the system is also used in order to perform a SMOW-SLAP calibration. How are evaporation/fractionation effects in the standards' vials handled? Based on Figure 3 we see no precautions concerning this.

Thank you for this comment. Indeed, this was not mentioned in the manuscript. Fractionation effects have been tested by sampling the water from the flasks every 2 weeks. We observed a maximum evolution of the isotopic composition of the standard in the bottle by 0.05 ‰ for $\delta^{18}$O and 0.5 ‰ in $\delta$D within 2 months in case of intense use of the instrument. This is now mentioned in the new version of the manuscript.

"The water in the water reservoirs is sampled every month to check its isotopic composition and renewed when the level of water is below half the maximum level. A maximum evolution of the isotopic composition of the lab-standard filling the water reservoirs has been observed as 0.05‰ and 0.5‰ respectively for $\delta^{18}$O and $\delta$D over a 2-month period. "

2.4 Evaluation–Measurement Stability and precision
The manuscript lacks important information on the evaluation of the performance of the system. Section 3.1 assures the reader that no fractionation is observed during the

generation of the vapour and its transfer in the flow lines while the agreement with the commercial standard delivery module is within 0.2 and 1‰ for $_{18}O$ and $_{D}$ respectively. It is almost impossible to judge these discrepancies based on figure 5 and in fact when looking closely it is rather difficult to see how the differences between the two systems are of this order.

With respect to figure 4 and the evaluation of the precision for [H2O] and $_{18}O/_{D}$ I strongly believe that the manuscript should include a proper Allan variance test (Werle et al , 2011; Steig et al , 2014). It is obvious that the system experiences drifts, whose origing is the vapour generator. Calculating the standard deviation on a 10 min window of a time series that obviously has a changing mean value looks and feels like possible cherry-picking.

2.4.1 Points to consider–Suggestions
• Show better evidence of zero fractionation. If you choose the SDM as a measure of comparison show a detailed comparison and plot the results clearly. The axes of fig 5 span 300 ‰ for $_D$. It is impossible to say anything.

A focus on the comparison between the SDM and the low-humidity level generator is now displayed on Figure S1:

[Figure]

*Figure S1*: *Comparison of the difference in isotopic composition ($\delta D$ top, $\delta^{18}O$ bottom) between the measured and the reference (measurement performed at 2000 ppmv) values as obtained with the SDM (red) and with the LHLG (black) coupled to the same Picarro L 2130-i with the same lab-standards (FP5) calibrated against VSMOW. The same measured $\delta^{18}O$ and $\delta D$ values are obtained at 2,000 ppmv through the SDM and the LHLG set-up.*

• Calculate the Allan variance of the system for 4-5 [H2O] levels. Let one water run for several hours and so you get a more honest idea about precision and drifts for all three parameters under consideration. It is a very simple experiment that can provide a valuable insight and by looking in the bibliography it is a rather standard

evalluation tool for laser spectroscopy based measurements.

Thank you for the suggestion. Indeed, this is a good idea and we have included the figure 3 (see answer to reviewer 1) in the revised manuscript

2.5 Water concentration correction

The water concentration correction is described in section 3.3 with Figure 5 being the main source of information for this part of the analysis. I would consider this one of the most important sections in the paper and unfortunately it spans only one paragraph. It is repeated in the manuscript that the system presented here is superior to the commercial standard delivery module below the level of 500 ppm. How can we see this when the SDM measurements in this are of $[H_2O]$ are not presented (measurements stop at $[H_2O]\_2000$ ppm)?

It is very difficult to get stable humidity plateau with a SDM below 500 ppmv. This is the reason why we could not display them. However, we include the following figure S2 (in addition to figure S1 shown in the previous answer to comment) in the new manuscript to strengthen the comparison between SDM and LHLG.

[Figure]

[Figure]

*Figure S2*: *Comparison of humidity plateaus (800 ppmv) generated with the LHLG (a) and with the SDM (b). The grey rectangles indicate period with only dry air injected.*

A text has also been added:

"The performance of the present LHLG can be compared to the performance of the SDM (see Supplementary Figures S1 and S2). First (Figure S2), a comparison has been performed at a humidity level of 800 ppmv, for which we have numerous daily calibrations performed with a SDM from a 4.5 years field deployment in Svalbard (Leroy-Dos Santos et al., 2020). The best SDM performance displays a standard deviation $1\sigma$ of 31 ppmv, which is significantly worse than the performance of the LHLG (standard deviation $1\sigma$ lower than 10 ppmv on average and down to 2 ppmv for 30% of the generated humidity plateaus). Second (Figure S1), while we measure the same influence of humidity on measured $\delta^{18}O$ and $\delta D$ either with the SDM or with the LHLG, the $1\sigma$ values on humidity levels are much larger for the SDM than for the LHLG."

The dataset in Figure 5 presents the difference of the raw isotopic value and a value that the authors call "reference" and "real". These words commonly refer to very specific things in isotope geochemistry and my guess is that the reference value is a SMOW-SLAP calibrated value (?). The term SMOW-SLAP calibration is not to be found at all in the manuscript. It is to some extent problematic that a water concentration correction measurement compares a raw value at a number of [H2O] levels with a value post SMOW-SLAP calibration. Please see my comments on SMOW-SLAP in the next section.

As mentioned above, the SMOW-SLAP calibration is now addressed in the revised manuscript and all our lab-standards are calibrated vs V-SMOW (using also V-SLAP as a reference).

As an example:
"First, the isotopic composition of three different lab-standards calibrated against VSMOW at LSCE ($H_2O$-$CO_2$ equilibration followed by IRMS for $\delta^{18}O$; Cavity RingDown Spectroscopy for $\delta D$; calibrated every 3 years using VSMOW and VSLAP provided by IAEA) have been compared, after their

generation by the present LHLG and by the commercial SDM, both at a humidity of 2,000 ppmv over 50-min time spans."

The type of correction is not described in the manuscript. This is what this work is all about. There can be various approaches on how it can be done (see for example (Schmidt et al , 2010; Gkinis et al , 2010; Aemisegger et al , 2012)) but the authors need to be open and specific about what they did exactly. Moreover (Schmidt et al , 2010) suggest that the response to changing [$H_2O$] levels varies with the isotopic composition. I do not know how important this effect is for this present work, but I believe that the authors should perform two separate experiments with two different waters (technically the system offers this possibility). It would also strengthen their claims for a vary stable system if they show the full experiment with its raw data as a function of time and not only the averages.

There is also a claim in this section that the stability of the system allows via the daily measurement of standards waters to quantify instrumental drifts. There is no strict evaluation of the stability of the system so far and I think that a proper Allan variance test at various levels of [$H_2O$] is necessary. But even in the absence of an Allan variance test the data given in Table 2 do not necessarily indicate instrumental drifts. All the isotope data in the table are within 1 ⍰ _ of the noise level for a 10-min period mentioned in the caption and there does not seem to be a one way trend. So how is it possible to say anything about instrumental drifts? Also the noise levels mentioned in the caption of Table 2 are very different for roughly the same [$H_2O$] when one looks at the data of Figure 4 (4.5 versus 14).

This is a good remark indeed and we are sorry for the confusion. Actually, while the noise on the humidity level is mainly due to the low-humidity level generator, the noise on the isotopic ratios depends on the laser analyzer which explains some inconsistency in the results which were displayed (we presented results obtained with different instruments). The inconsistencies will be fixed in the revised version. Indeed, Table 2 shows that there is no measurable drift and this is now explained in the revised version.

"The stability of the LHLG allows a robust quantification of the L2130-i analyzer drift thanks to a daily measurement of the same water isotopic standard reference (see Table S3 showing actually no measurable drift over a 3-week period)."

2.5.1 Points to consider-Suggestions
• A clear description of the correction is missing. Some math is needed here.

We provide the following explanation in the revised manuscript.

Our data show a result already observed in Weng et al. (2020): while the dependency of $\delta^{18}O$ and $\delta D$ to humidity is similar for low $\delta^{18}O$ and $\delta D$ lab-standards (NEEM and FP5), we observe a different behavior for the $\delta D$ vs humidity relationship for the high $\delta^{18}O$ and $\delta D$ lab-standard EPB. This result strengthens the recommendation of Weng et al. (2020) to use two water standards in the range of the measured water vapor isotopic composition to best calibrate our final data. In our case, our applications were in Antarctica, so that we used our two lowest lab-standards (NEEM and FP5). For the two standards and for this particular Picarro L2130-i (results are expected to depend on the instrument), the same dependency of isotopic composition vs humidity is observed. We express this dependency as the relationship between the difference in $\delta D$ or $\delta^{18}O$ between the measured value at the given humidity and the value of the same standard measured at a humidity of 2,000 ppmv. The

experimental data for NEEM and FP5 from Figure 5 are fitted through polynomial functions with respect to humidity h (in ppmv):

$\delta^{18}O - \delta^{18}O_{ref} = 3.97 \times 10^{-18} \times h^6 - 3.59 \times 10^{-14} \times h^5 + 1.28 \times 10^{-10} \times h^4 - 2.31 \times 10^{-7} \times h^3 + 2.19 \times 10^{-4} \times h^2 - 1.06 \times 10^{-1} \times h + 23.7$ (eq.1)

$\delta D - \delta D_{ref} = 6.86 \times 10^{-17} \times h^6 - 6.00 \times 10^{-13} \times h^5 + 2.08 \times 10^{-9} \times h^4 - 3.61.10^{-6} \times h^3 + 3.31.10^{-3} \times h^2 - 1.54 \times h + 313$ (eq. 2)

After this correction, the measured values corrected from humidity dependence are corrected using the comparison of the measured values of the 2 standards at 2,000 ppmv to their VSMOW calibrated values as explained in section 3.5 below."

• An experiment at two different isotopic levels showing the raw data versus time will show if there is an isotope effect in the water concentration dependence.

We think that new figures 3 and 5 answer exactly this comment.

• A proper quantification of instrumental drifts (and this concerns the system as a whole and not only the spectrometer) can be done with a proper Allan variance test.

This is now shown on figure 3 which is implemented in the revised version of the manuscript.

2.6 Lack of reference to SMOW-SLAP calibration
The main goal of building a water vapour generator as a peripheral for isotope measurements of water vapour is to be able to calibrate the dataset on the SMOW-SLAP scale. This is the only way to communicate and compare the measurements with other existing data sets and produce some science out of them. It is also even more important if the deuterium excess parameter will be studied as it is very sensitive to this calibration procedure. Therefore it appears very awkward that a manuscript dealing with this topic does not include a single note, comment or reference to this very important step Of the measurement process. The dataset presented later in section 4 of the manuscript are impossible to evaluate if they are not calibrated in the SMOW-SLAP scale.
One more purpose of performing such calibrations, is that they can reveal possible accuracy issues in the instrumentation system. Given two standard waters one should be able to produce a calibration line and thereafter measure a third water of known isotopic composition treating it as an unknown. If the resulting value lies beyond the 3 ꟍ _ range then there is likely something wrong with the system. That could be any part from the water standard storage to the water vapour generation system or the spectrometer itself. Currently there is no way to say anything about the accuracy of the system. With this in mind, section 4 of the manuscript is of very little use as the dataset is reported on some local instrument scale.
A SMOW-SLAP calibration experiment at various [H2O] levels using the SDM and the current system would provide a proper comparison between the two systems and therefore it would be a very important addition to the manuscript.

2.6.1 Points to consider
• A proper treatment of the SMOW-SLAP calibration step is notably missing.

All our lab-standards are regularly calibrated vs SMOW using IAEA VSMOW and VSLAP standards (every 3 years in minimum). In order to calibrate our lab-standards, we use determination from

Picarro L 2130-i for $\delta D$ and for $\delta^{18}O$. In addition, we use a $\delta^{18}O$ calibration using an IRMS through water – $CO_2$ equilibration.
New text:

"First, the isotopic composition of three different lab-standards calibrated against VSMOW at LSCE ($H_2O$-$CO_2$ equilibration followed by IRMS for $\delta^{18}O$; Cavity RingDown Spectroscopy for $\delta D$) have been compared, after their generation by the present LHLG and by the commercial SDM, both at a humidity of 2,000 ppmv over 50-min time spans."

• Performing 2-standard calibrations and measuring a third water standard treated as an unknown will be a valuable -almost essential- addition to the manuscript, offering important information on the accuracy of the system.

This is an excellent suggestion and this test has been performed and a new section 3.5 has been added:

**3.5- Accuracy of the system**

The accuracy of the system has been addressed performing a 2-standard calibration and measuring a third standard treated as an unknown. We used two lab-standards calibrated vs VSMOW with large $\delta^{18}O$ and $\delta D$ differences (EPB and FP5) and used the lab-standard NEEM, also independently calibrated against VSMOW. The 3 lab-standards have been vaporized at 800 ppmv and measured by the same L2130-i analyzer.

| Standard | VSMOW calibrated value | Measured value at 800 ppmv | Measured value corrected from humidity dependence (Equation 1) |
| --- | --- | --- | --- |
| EPB | -6.24 ‰ | -8.27 ‰ | -10.78 ‰ |
| NEEM | -33.5 ‰ | -34.48 ‰ | -36.99 ‰ |
| FP5 | -48.33 ‰ | -49.02 ‰ | -51.53 ‰ |

***Table 3****: Comparison of measured vs VSMOW calibrated $\delta^{18}O$ values for 3 standards measured with a Picarro analyzer after generation of water vapor using the LHLG.*

We used the measured and true values of EPB and FP5 to estimate the $\delta^{18}O$ value of the NEEM standard from its measured value (Table 3). Using the linear relationship obtained from VSMOW calibrated EPB and FP5 $\delta^{18}O$ vs measured EPB and FP5 $\delta^{18}O$ values following the recommendations of the National Institute of Standards and Technology (NIST, reference material 8535a) leads to an estimated NEEM $\delta^{18}O$ of -33.31 ‰ to be compared to the independently VSMOW calibrated value of -33.5 ‰. Given the uncertainty of about 0.8-1 ‰ when measuring $\delta^{18}O$ around 800 ppmv, we can conclude that the system is accurate.

• Since a lot has been written about the performance of the commercial SDM it will be proper to perform 2-standard calibrations for various [$H_2O$] levels and compare the results.

See answer above as well as the Figures S1 and S2 added to answer this comment.

**References:**

Kerstel, E.: Modeling the dynamic behavior of a droplet evaporation device for the delivery of isotopically calibrated low-humidity water vapor, Atmos. Meas. Tech. Discuss. [preprint], https://doi.org/10.5194/amt-2020-428, in review, 2020.

Landsberg, J.: Développement d'un spectromètre laser OF-CEAS pour les mesures des isotopes de la vapeur d'eau aux concentrations de l'eau basses. [online] Available from: http://www.theses.fr/2014GRENY052/document, 2014.

Merlivat, L., Molecular diffusivities of $H_2^{16}O$, $HD^{16}O$ and $H_2^{18}O$ in gases, J. Phys. Chem., 69(6), 2864–2871, 1978.

---

## Referee Report (RR1)

**Review: A dedicated robust instrument for water vapor generation at low humidity for use with a laser water isotope analyzer in cold and dry polar regions." by Christophe Leroy-Dos Santos et al. - Revised version**

**Very satisfactory revision and a few remarks for further improvement**

This is the first iteration of the manuscript by Santos et al. The authors have indeed provided very satisfactory answers to all the questions/remarks raised and the modifications they have performed have improved the manuscript considerably. Thereby I would recommend the manuscript for publication in AMT after the following four points are addressed. I would also recommend a final "scanning" of the manuscript for possible language use glitches.

**1 Allan variance tests**

The authors have performed a very useful Allan variance test. The results of the test however are not discussed thoroughly and the test could be described/introduced a litte bit better in the opening of section 3.2.

Some readers may not be familiar with the term Allan variance and how such a test is performed and data-processed. Please provide some information on

the aspects of your test that would help the reader learn more about your methods and not only your results.

From the plots of the results there emerge some interesting findings. Firstly the stability of your water concentation signal seems to be much worse when compared to the $\delta^{18}$O signal. However this does not seem to affect your ability to improve your signal-to-noise in the $\delta^{18}$O signal by further averaging your measurements. In other words to say that your isotope signal depends strongly on the humidity level is not exactly right. Another important finding is that for averaging times around 500 s the signal-to-noise seems to be very comparable between all humidity levels. This is also important to note.

I would encourage the authors to look closely at the results of the Allan variance test and commend carefully on them. Suggestions for optimal averaging times can be drawn. Interestingly enough the $\delta^{18}$O  and $\delta$Dsignal at the level of 320 ppm seem to show the best performance with respect to stability and potential for improving the signal-to-noise through averaging. It could also be a random result. Whatever the case I would recommend utilising this test even more for the further development of the system and dare to go even lower in concentration (100, 200 ppm) to see if this stability is observed there. These further tests would be extremely welcome as an addition to the current revised version but I would not require them for publication.

**2   Effect of air flow and Table S2**

A description of how this test is performed and the results of the test itself belong in the main part of the manuscript. Please provide more text introducing the experiment and the methodology used.

**3 Humidity correction equation**

The equation the authors provide is a high order polynomial, which from my experience is prone to overfitting and erroneous behavior at the edges of the fitted interval. I would appreciate a Figure 5 that is big in size with the correction curve included in the plot. Discuss edge effects using the polynomial correction (if any) and define limits if such effects are apparent.

**4 VSMOW-SLAP**

Section 3.5 is too short and in my view should have a title that refer to the term SMOW-SLAP calibration. I would also like to see a calibration curve where the two extreme points (EPB, FP5) are used to define the line of calibration and NEEM is treated as an unknown. Even though to some they may appear trivial it is good practice if you write the equations on the slope and intercept of the calibration line. It also needs to be stressed out that the use of the standard waters and the calibration procedure is not there only for the assessment of the accuracy but it is a vital part of the post processing of the vapour measurements. The term VSMOW is not found in neither the abstract nor the introduction of the manuscript. It needs to be mentioned so the reader knows that the authors have addressed the essential step of the SMOW-SLAP calibration.

---

## Author Response (AR2)

**Very satisfactory revision and a few remarks for further improvement**

This is the first iteration of the manuscript by Santos et al. The authors have indeed provided very satisfactory answers to all the questions/remarks raised and the modifications they have performed have improved the manuscript considerably. Thereby I would recommend the manuscript for publication in AMT after the following four points are addressed. I would also recommend a final "scanning" of the manuscript for possible language use glitches.

**1 Allan variance tests**

The authors have performed a very useful Allan variance test. The results of the test however are not discussed thoroughly and the test could be described/introduced a litte bit better in the opening of section 3.2.

Some readers may not be familiar with the term Allan variance and how such a test is performed and data-processed. Please provide some information on the aspects of your test that would help the reader learn more about your methods and not only your results.

From the plots of the results there emerge some interesting findings. Firstly the stability of your water concentation signal seems to be much worse when compared to the  $\delta$ 18O signal. However this does not seem to affect your ability to improve your signal-to-noise in the  $\delta$ 18O signal by further averaging your measurements. In other words to say that your isotope signal depends strongly on the humidity level is not exactly right. Another important finding is that for averaging times around 500 s the signal-to-noise seems to be very comparable between all humidity levels. This is also important to note.

I would encourage the authors to look closely at the results of the Allan variance test and commend carefully on them. Suggestions for optimal averaging times can be drawn. Interestingly enough the  $\delta$ 18O and  $\delta$ Dsignal at the level of 320 ppm seem to show the best performance with respect to stability and potential for improving the signal-to-noise through averaging. It could also be a random result. Whatever the case I would recommend utilising this test even more for the further development of the system and dare to go even lower in concentration (100, 200 ppm) to see if this stability is observed there. These further tests would be extremely welcome as an addition to the current revised version but I would not require them for publication.

>> Indeed, this part could be improved and we followed the advice of the reviewer. First, we present now two additional Allan variances obtained at 170 ppmv with 2 different stability results for this humidity level: in one case the Allan variance is very stable (yellow curve on new figure 3) and in the other case, the minimum Allan variance is significantly higher while still under 10 ppmv (blue curve on new figure 3). For both cases, the minimum Allan variance for  $\delta^{18}$ O and  $\delta$ D is significantly higher than the one obtained at higher humidity level (black and blue curves on new figure 3).

We have inserted this new figure in the revised manuscript with a text to follow the reviewer advice and better explain the Allan variance concept and how we use it:

**Figure 3:** Allan deviation over 4 hours for different humidity levels (black 1,080 ppmv; dark blue 770 ppmv; green 400 ppmv; pink 320 ppmv; yellow and light blue 170 ppmv) for humidity (a),  $\delta^{18}$ O (b) and  $\delta$ D (c).

A proper approach to quantify the stability of our system is to use the Allan variance defined as:

$$\sigma_{y}^{2}(t) = \frac{1}{2}(\langle y_{n+1} - y_{n} \rangle^{2})$$
 (eq. 3)

where  $y_n$  are the successive measurements over a period t.

An Allan variance plot as a function of averaging time is indeed useful to determine the optimal time over which the sample humidity and the isotopic composition should be averaged to obtain a precise determination (low standard deviation) and avoid drift. Figure 3 displays the Allan deviation (square root of the Allan variance) in  $\delta^{18}$ O,  $\delta$ D and humidity obtained by running a long plateau of standard A or standard B in the "infuse" mode over 4 hours for different humidity levels. The humidity variance always stays below 10 ppmv over the 4 hours test and the  $\delta^{18}$ O and  $\delta$ D Allan deviations display minimum values below 1 ‰ and 7 ‰ respectively. The minimum value for the  $\delta^{18}$ O and  $\delta$ D Allan deviation of  $\delta^{18}$ O and  $\delta$ D is dependent on the analyzer used, we observe that the Allan deviation at 1000 s (17 minutes) for  $\delta^{18}$ O and  $\delta$ D also depends to some extent on the humidity level: the lowest levels are obtained for humidity levels of 770-1,080 ppmv and the highest levels are obtained for humidity level

2 Effect of air flow and Table S2

A description of how this test is performed and the results of the test itself belong in the main part of the manuscript. Please provide more text introducing the experiment and the methodology used.

>> The Table S2 has been placed in the main text and an explanation has also been written as follows:

"The stability of the instrument has been tested over a large range of parameters. We show an example in Table 2. We modified the air flow associated with standard A (the same results can be obtained with standard B) between 200 and 400 sccm with an air flow on channel B of half the value of channel A. The infusion rate was varied between 0.03 and 0.14  $\mu$ L/min in order to produce humidity levels of 400 and 800 ppmv. The 1 $\sigma$  standard deviations observed over 10 minutes plateaus are comparable to the standard deviation obtained when the air flow is set to 300 sccm."

3 Humidity correction equation

The equation the authors provide is a high order polynomial, which from my experience is prone to overfitting and erroneous behavior at the edges of the fitted interval. I would appreciate a Figure 5 that is big in size with the correction curve included in the plot. Discuss edge effects using the polynomial correction (if any) and define limits if such effects are apparent.

>> We followed this suggestion with the new figure 5 displayed below. We also added a sentence to specify the range of validity of equations 4 and 5, see below: